# CD8 lymphocytes mitigate HIV-1 persistence in lymph node follicular helper T cells during hyperacute-treated infection

Omolara O. Baiyegunhi [1,2], Jaclyn Mann[2], Trevor Khaba[2], Thandeka Nkosi[1], Anele Mbatha[2], Funsho Ogunshola[1,3], Caroline Chasara[1], Nasreen Ismail[2], Thandekile Ngubane[2], Ismail Jajbhay[4], Johan Pansegrouw[4], Krista L. Dong[3], Bruce D. Walker [1,2,3,5,6,7], Thumbi Ndung'u[1,2,3,8,9] & Zaza M. Ndhlovu [1,2,3 ✉]

HIV persistence in tissue sites despite ART is a major barrier to HIV cure. Detailed studies of HIV-infected cells and immune responses in native lymph node tissue environment is critical for gaining insight into immune mechanisms impacting HIV persistence and clearance in tissue sanctuary sites. We compared HIV persistence and HIV-specific T cell responses in lymph node biopsies obtained from 14 individuals who initiated therapy in Fiebig stages I/II, 5 persons treated in Fiebig stages III-V and 17 late treated individuals who initiated ART in Fiebig VI and beyond. Using multicolor immunofluorescence staining and in situ hybridization, we detect HIV RNA and/or protein in 12 of 14 Fiebig I/II treated persons on suppressive therapy for 1 to 55 months, and in late treated persons with persistent antigens. CXCR3[+] T follicular helper cells harbor the greatest amounts of *gag* mRNA transcripts. Notably, HIV-specific CD8[+] T cells responses are associated with lower HIV antigen burden, suggesting that these responses may contribute to HIV suppression in lymph nodes during therapy. These results reveal HIV persistence despite the initiation of ART in hyperacute infection and highlight the contribution of virus-specific responses to HIV suppression in tissue sanctuaries during suppressive ART.

[1] Africa Health Research Institute (AHRI), Durban, South Africa. [2] HIV Pathogenesis Programme, The Doris Duke Medical Research Institute, University of KwaZulu-Natal, Durban, South Africa. [3] Ragon Institute of Massachusetts General Hospital, Massachusetts Institute of Technology, and Harvard University, Cambridge, MA, USA. [4] Prince Mshiyeni Memorial Hospital, Durban, South Africa. [5] Institute for Medical Sciences and Engineering and Department of Biology, Massachusetts Institute of Technology, Cambridge, MA, USA. [6] Howard Hughes Medical Institute, Chevy Chase, MD, USA. [7] Department of Immunology, Harvard Medical School, Boston, MA, USA. [8] Max Planck Institute for Infection Biology, Berlin, Germany. [9] Division of Infection and Immunity, University College London, London, UK. ✉email: zndhlovu@mgh.harvard.edu

Antiretroviral therapy (ART) does not eradicate HIV infection due in large part to early establishment and persistence of integrated proviruses in quiescent circulating and tissue reservoirs[1–3], which are resistant to drug or immune-mediated clearance. Additional mechanisms of persistence that involve ongoing replication have been suggested[4], including replication in germinal centers (GCs) within secondary lymphoid tissues, which have been described as sanctuary sites due to low CD8+ T cell infiltration[5] and suboptimal penetration of antiretroviral drugs[6]. Ongoing virus replication, and to a lesser extent viral gene expression in the face of ART are contested concepts, with studies of individuals initiated on ART during chronic infection yielding conflicting results[6–12]. T follicular helper (Tfh) cells have been identified as a major source of persistent virus[13], but the precise subset of these cells enriched for HIV transcription on therapy within sanctuary sites is unknown. HIV persistence on ART is underscored by the nearly inevitable rebound of plasma viremia when treatment is interrupted, even after years of suppression, and is the major barrier to HIV cure[14].

It has previously been shown that very early initiation of ART can lead to prolonged remission when treatment is interrupted[15,16], but this is an infrequent occurrence. In most individuals, virus rebound occurs within weeks to months even in individuals initiated on ART during Fiebig stage I (hyperacute) HIV infection[14,17], despite rapid suppression of viremia and dramatically lower numbers of latently infected cells in peripheral blood (PB)[18–21]. Intriguingly, virus rebound kinetics following treatment interruption are heterogenous, sometimes taking a year or more[14,22]. The underlying immunological and virologic mechanisms responsible for the diverse viral rebound kinetics remain unknown. For instance, prolonged therapy in SIV infected macaques initiating therapy within 6 days of infection, prior to detectable plasma viremia, led to apparent elimination of infection after 600 days of suppressive ART in some animals[23]. However, such clearance of infection has not been observed in acute HIV infection, even in persons in whom therapy was initiated before detection of plasma viremia[20]. We previously showed that early treatment initiation enhances T cell functions in PB and limits viral diversity[24], but it is not clear if functional responses occur in tissues and whether such responses play a significant role in HIV suppression during ART. Limited access to tissue samples from persons initiating therapy before peak viremia has impeded a better understanding of the impact of early therapy on the lymphoid reservoir.

Here we analyzed 64 excisional lymph node (LN) biopsies and paired PB samples obtained from a well pedigreed cohort of individuals, where some initiated ART during hyperacute HIV infection. We investigated the impact of blunting peak viremia on the microanatomical location, cellular source and role of T cell responses on HIV persistence in LNs. Study participants were drawn from a unique hyperacute HIV infection cohort termed Females Rising through Education, Support and Health (FRESH). FRESH is a prospective study of uninfected 18–23-year-old women at high risk of HIV infection established at the epicenter of the HIV epidemic in South Africa, where yearly incidence rates approach 10%. The participants were offered PrEP when it

became available in South Africa. Uptake was very high (90%) but retention was poor similar to other PrEP programs in the region. Despite vigorous prevention efforts, twice weekly monitoring for viral RNA has identified and treated (Tx) persons at the onset of plasma viremia, allowing for immediate institution of ART in many cases resulting in peak plasma viral loads that are sometimes <1000 RNA copies/ml and the preservation of CD4+ T cell numbers[18]. Our results show that despite ART-induced blunting of peak viremia[18] and augmentation of functional HIV-specific T cell responses[24], HIV Gag p24 protein and viral RNA can persist in the LNs of Fiebig I/II Tx donors even after 4.5 years of fully suppressive ART, and these viral antigens are enriched in LN CXCR3+Tfh cells. We also show that superior functioning T cell responses were associated with lower HIV antigen persistence in the LNs.

## Results

**Hyperacute HIV infection as a model to interrogate antigen persistence in lymph nodes.** To determine the impact of immediate initiation of ART in hyperacute HIV infection (before peak viremia) on HIV clearance from sanctuary sites, we studied 14 women aged 18–26 who initiated ART during hyperacute HIV infection (Fiebig I/II Tx) and achieved full suppression of plasma viremia within a median of 15 days (range, 6–33). LNs were obtained by excisional biopsy after treatment for a median of 370 days (range, 19–1647). All remained fully suppressed except for one donor who had a transient viral load blip prior to LN excision. Five additional individuals identified in Fiebig stages III–V of infection and started on ART 1 day after diagnosis were also included. Three additional control groups were included: 13 HIV negative (HIVneg) donors; 17 individuals who initiated treatment in Fiebig VI and beyond (late Tx); and 15 untreated individuals whose duration on infection is unknown (unTx). Detailed characteristics of the cohorts are in Table 1. In total, 95% of the study participants were females.

**Long-term persistence of HIV Gag p24 antigen in germinal centers (GCs) of individuals initiating antiretroviral therapy during hyperacute HIV-infection.** To investigate HIV persistence in LNs of individuals initiating ART in Fiebig stages I/II, we measured HIV Gag p24 antigen in excisional LN biopsies by multicolor immunofluorescence (IF) staining of formalin-fixed paraffin-embedded LNs and imaging of tissue sections (Supplementary Fig. 1). The transcription factor BCL-6 was used to identify active GCs[25] (Supplementary Fig. 1a–d) and images were quantified for Gag p24 content using the algorithm for area measurements in TissueQuest (TissueGnostics)[26]. Figure 1a shows a representative image of HIV Gag p24 LN staining for a participant who was diagnosed in Fiebig stage I, initiated ART within 48 h and achieved persistent plasma viremia suppression within 33 days. The LN sample shown was obtained after 479 days of uninterrupted ART treatment with undetectable viremia, and depicts HIV Gag p24 antigen within a GC, which

**Table 1 Characteristics of study participants at the time of lymph node excision.**

| Characteristic | Fiebig stage I/II treated | Fiebig stage III–V treated | Late treated[a] | Untreated | HIV negative |
|---|---|---|---|---|---|
| No. of participants (% female) | 14 (100%) | 5 (100%) | 17 (94%) | 15 (80%) | 13 (100%) |
| Median (IQR) age of participants (years) | 22 (20–24) | 22 (19–24) | 24 (22–27) | 24 (24–29) | 22 (21–23) |
| Median (IQR) CD4 Count (no. of cells/μl) | 911 (717–1120) | 942 (696–1086) | 589 (454–792) | 595 (401–720) | 976 (792–1180) |
| Median (IQR) plasma viral load copies/ml | <20 (<20–<20) | <20 (<20–<20) | <20 (<20–<20) | 11,000 (1900–22,000) | NA |
| Median (IQR) days on treatment | 370 (31–550) | 120 (46–677) | 571 (100–762) | NA | NA |

*NA* not applicable, *IQR* interquartile range.
[a]Donors whose Fiebig stage of infection was either Fiebig VI or unknown are defined as late treated.

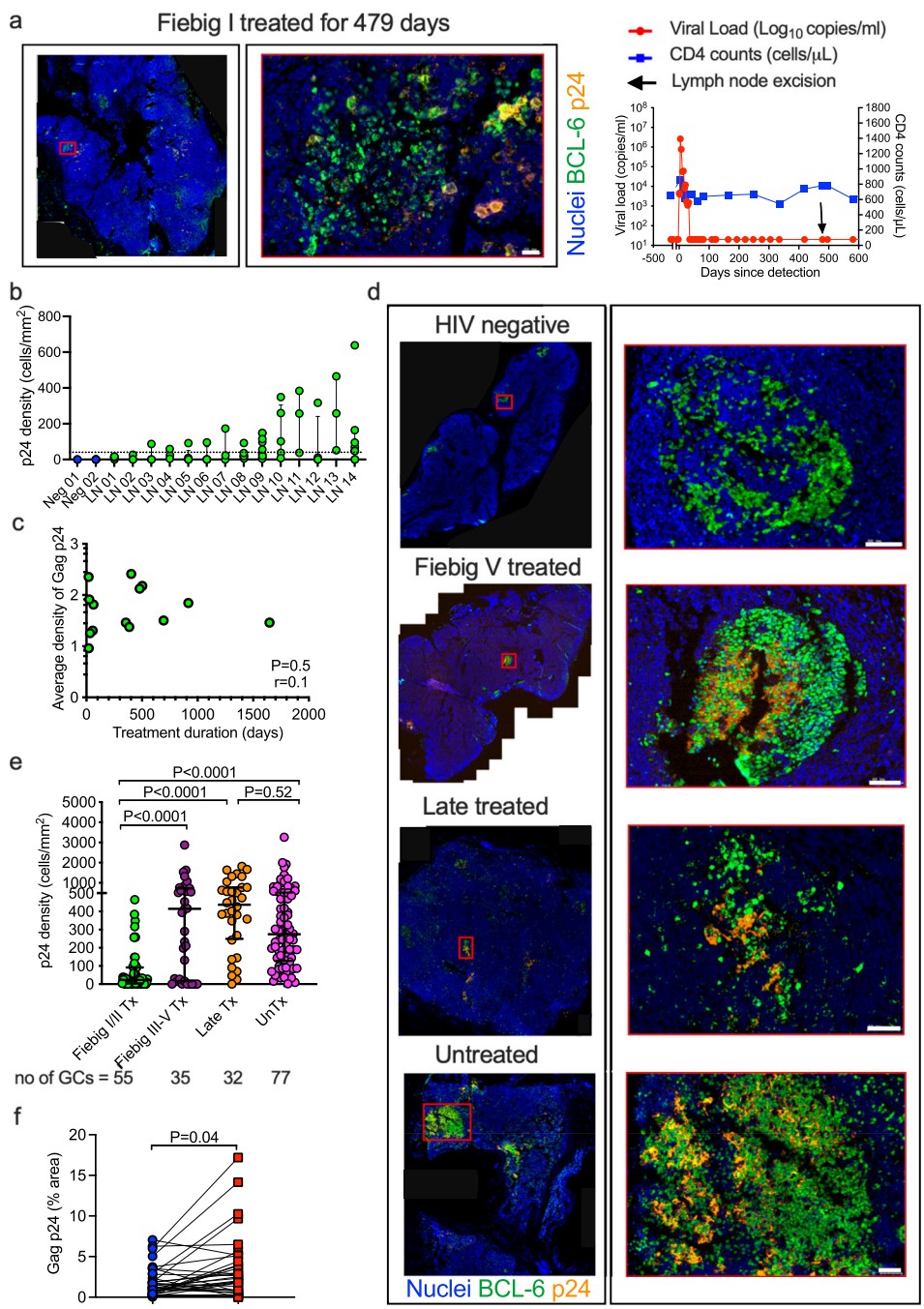

**Fig. 1 HIV Gag p24 persists in germinal centers (GCs) despite early ART initiated in Fiebig stages I/II. a** IF images of Gag p24 antigen (yellow) and BCL-6 (green) staining in lymph node (LN) sections and a scatter plot showing CD4+ T cell counts, plasma viral loads and the time of LN excision for a Fiebig I treated (Tx) participant. Nuclei are counterstained with DAPI (blue) and scale bar is 20 μm. **b** Gag p24 density per GC computed from TissueQuest (TissueGnostics, Vienna) analysis of IF LN sections (HIV negative, $n = 2$; Fiebig I/II Tx, $n = 14$), and **c** correlation analyses of average density of Gag p24 per donor and the treatment duration prior to LN excision. **d** Representative IF images of Gag p24 antigen (yellow) and BCL-6 (green) staining (scale bar is 50 μm) and **e** aggregate data from TissueQuest (TissueGnostics, Vienna) analysis of IF LN sections (Fiebig I/II Tx, $n = 14$; Fiebig III–V Tx, $n = 3$; late Tx, $n = 8$; and untreated (unTx), $n = 13$). Each dot represents the density of Gag p24 per GC and the total number of GCs analyzed per group is displayed. **f** Comparison of the total percentage area staining for Gag p24 within GCs and outside the GCs (EF) for all donors ($n = 36$). Three independent experiments were conducted with similar results. Error bars represent interquartile range. Data are presented as median ± interquartile range. All statistical tests are two-sided. Two-tailed $p$ values from Mann–Whitney $U$ test (**f**) or adjusted $p$ values from Dunn's multiple comparison's test (**e**) are shown. Spearman rho ($r$) values and $p$ values are reported for correlation analyses (**c**). Dotted line denotes threshold of detection (**b**). Source data are provided as a source data file.

was present in 5 of the 7 GCs examined in this LN (Supplementary Fig. 1e).

Gag p24 staining and imaging were conducted on a total of 14 LNs from fully suppressed Fiebig I/II Tx donors obtained at a median of 370 days (range, 19–1647 days) post-ART initiation. Twelve of 14 (86%) of these these Fiebig I/II Tx donors had detectable HIV Gag p24 in at least one GC, and overall 42 of 55 GCs evaluated (76%) were positive for Gag p24. In two Fiebig I/II Tx donors there was no detectable HIV Gag p24 despite examining more than more than four GCs (Fig. 1b). In those with detectable HIV Gag p24, quantitative image analysis revealed no correlation between the amount of HIV Gag p24 present in the LN tissue section and treatment duration prior to LN excision (Fig. 1c). Notably, regardless of treatment duration, Fiebig I/II Tx donors had significantly less detectable HIV Gag p24 compared to Fiebig III–V Tx ($p < 0.0001$), late Tx ($p < 0.0001$) and unTx ($p < 0.0001$) donors, though there was considerable overlap (Fig. 1d, e and Supplementary Fig. 1f, g). This result was also consistent in a subset of donors Tx beyond 1 year (Supplementary Fig. 1h). Quantitative image analysis of all treated LNs revealed a trend of greater area percent of Gag p24 staining in some GCs compared to extrafollicular areas of the tissue ($p = 0.04$, Fig. 1f and Supplementary Fig. 1i). Together, these data demonstrate that early ART initiation in Fiebig stage I/II limits the magnitude of HIV Gag p24 antigen in LNs, but that Gag p24 can persist predominantly in follicular areas even after 4.5 years of fully suppressive treatment.

**Lymph nodes of Fiebig I/II treated individuals harbor HIV-1 RNA**. To determine if viral RNA transcription was occurring, which is required to produce infectious virions, we used an in situ hybridization (ISH) assay called RNAscope[27] to probe for HIV-1 gag-pol RNA within LN sections. 12 Fiebig I/II Tx, 4 Fiebig III–V Tx, 2 late Tx, 4 unTx and 3 HIVneg LN samples were analyzed based on sample availability. Viral RNA was detected as punctate dots in LNs from all HIV-infected persons and there were no signals in the HIVneg controls (Fig. 2a, b and Supplementary Fig. 2). Productively infected viral RNA$^+$ cells were identified as a dense spherical signal, whereas follicular dendritic cell (FDC)-bound virus particles were defined by a diffuse lattice-like pattern consistent with previous reports[28]. Combined RNAscope® ISH gag-pol staining with IF staining for CD4$^+$ T cells confirmed viral RNA (green) within CD4$^+$ T cells (red, Fig. 2c). RNAscope staining was quantified using Fiji[29]. Ten of 12 Fiebig I/II Tx donors had detectable but significantly lower amounts of HIV RNA compared to late Tx ($p = 0.04$) and unTx ($p = 0.001$) donors (Fig. 2d). However, there was no difference in RNA density between Fiebig I/II Tx and Fiebig III–V Tx donors. Notably, there was a positive correlation between Gag p24 density measured by IF and gag-pol RNA measured by in situ hybridization in Fiebig I/II Tx donors ($p = 0.002$; $r = 0.8$, Fig. 2e). The results are consistent with the persistence of viral RNA despite very early ART initiation in hyperacute infection and durable plasma virus suppression.

**Discordant HIV-1 RNA loads in plasma and lymph nodes**. To better define active virus transcription within LN mononuclear (LNMCs) cells and to determine the viral loads in the LNs of aviremic individuals initiated on treatment either very early or later in infection. We measured cell-associated viral loads in LNMCs using a commercial viral load assay Cobas® Ampliprep HIV-1 test. We found a hierarchy of LNMC viral loads with the values lowest in patients that initiated therapy in Fiebig stages I/II (Fig. 2f). Interestingly, neither the peak plasma viral load

(Fig. 2g), treatment duration before LN excision (Fig. 2h), nor the time to suppression (Fig. 2i) impacted viral RNA persistence in the LN. Overall, quantifiable amounts HIV RNA persists in the LNs of most Fiebig I/II Tx individuals and the magnitude of LN viral loads was not dependent on the duration of treatment.

**Expansion of GCTfh cells in early ART-treated individuals**. Identifying the cellular phenotypes of persistent HIV-1 protein and transcripts during therapy will be critical for future anti-HIV interventions. While follicular T helper (Tfh) cells are a key component of the adaptive immune response to HIV-1 infection and provide cognate help to B cells[30,31] and CD8$^+$ T cells[32,33], these cells also serve as a major HIV reservoir[13,34]. Moreover, HIV antigen can be trapped in the follicular dendritic lymphoreticular network within LNs and persist for years[35], thus we interrogated persistence within these cell subsets.

Given that Tfh are major targets of HIV infection, we first sought to determine the extent to which early ART mitigates HIV-induced Tfh expansion. We defined GCTfh as CD4$^+$CD45RA$^-$CXCR5$^{hi}$PD-1$^{hi}$ and nonGCTfh cells as CD4$^+$CD45RA$^-$CXCR5$^+$PD-1$^+$ in LNMCs (Fig. 3a and Supplementary Fig. 3a) consistent with previous Tfh studies[36]. HIV negatives ($n = 9$) had very low frequencies of GCTfh cells (median 1.3%, IQR; 0.6% to 1.5%) of antigen experienced (CD45RA$^-$) CD4$^+$ T cells whereas nonGCTfh cells were 11% (IQR; 8.5% to 13%, Fig. 3b). HIV infection resulted in significant expansion of GCTfh (Fig. 3c). Treatment initiation impacted the extent of GCTfh expansion. Immediate therapy was associated with significant diminution of GCTfh expansion (Fiebig I/II Tx vs. unTx $p = 0.002$), which was comparable among all treatment groups ($p =$ ns, Fig. 3c). Notably, HIV-induced Tfh expansion was restricted to GCTfh, as no significant expansion of nonGCTfh cells were observed between the groups (Fig. 3d). To verify these observations, we quantified the area of GCs and area densities of GCTfh in situ using FFPE LNs (Fig. 3e–g). Consistent with flow cytometry data GCs and GCTfh cell densities were expanded in HIV infection and significantly greater in unTx infection compared to Fiebig I/II Tx and HIV negative controls (Fig. 3f, g). The numbers of GCs per patient (Supplementary Fig. 3b, c, e) and the average area of GCs (Supplementary Fig. 3d, f) correlated with HIV p24 and RNA measurements in tissue (Supplementary Fig. 3c–f). Furthermore, comparisons of cell phenotype markers of activation or inflammation (Supplementary Fig. g–j) as well as the CD4$^+$/PD1$^+$ ratio of cells within GCs (Fig. S3k), among the treatment groups were inconclusive. Together, these data show that early treatment initiated in Fiebig I/II mitigates HIV-induced GCTfh expansion. Reduced HIV targets in GCs might partly explain reduced HIV persistence in LN of individuals who initiate therapy early.

To gain more insight on cellular targets of HIV infection in LNs, we investigated if there was a particular subset of GCTfh that was selectively expanded. We quantified previously described[37,38] GCTfh subsets namely; GCTfh1 defined as CXCR3$^+$CCR6$^-$, GCTfh2 as CXCR3$^-$CCR6$^-$, double positive (dp)GCTfh as CXCR3$^+$CCR6$^+$ and R6$^+$GCTfh defined as CXCR3$^-$CCR6$^+$ (Fig. 3h) among our study groups (Fig. 3i) and determined their relationship with HIV Gag p24 densities. While subsets had varying frequencies (Fig. 3i), within the Fiebig I/II and III–V Tx donors, there was a trend of higher frequency of GCTfh1 being associated with greater degrees of Gag p24 positivity ($p = 0.08$; $r = 0.5$) (Fig. 3j), whereas R6$^+$GCTfh displayed a weak negative association ($p = 0.08$; $r = 0.5$) (Fig. 3k). Overall, these results show that while early treatment mitigates GCTfh responses, subset distribution of Tfh cells might impact virus persistence in early treated LNs.

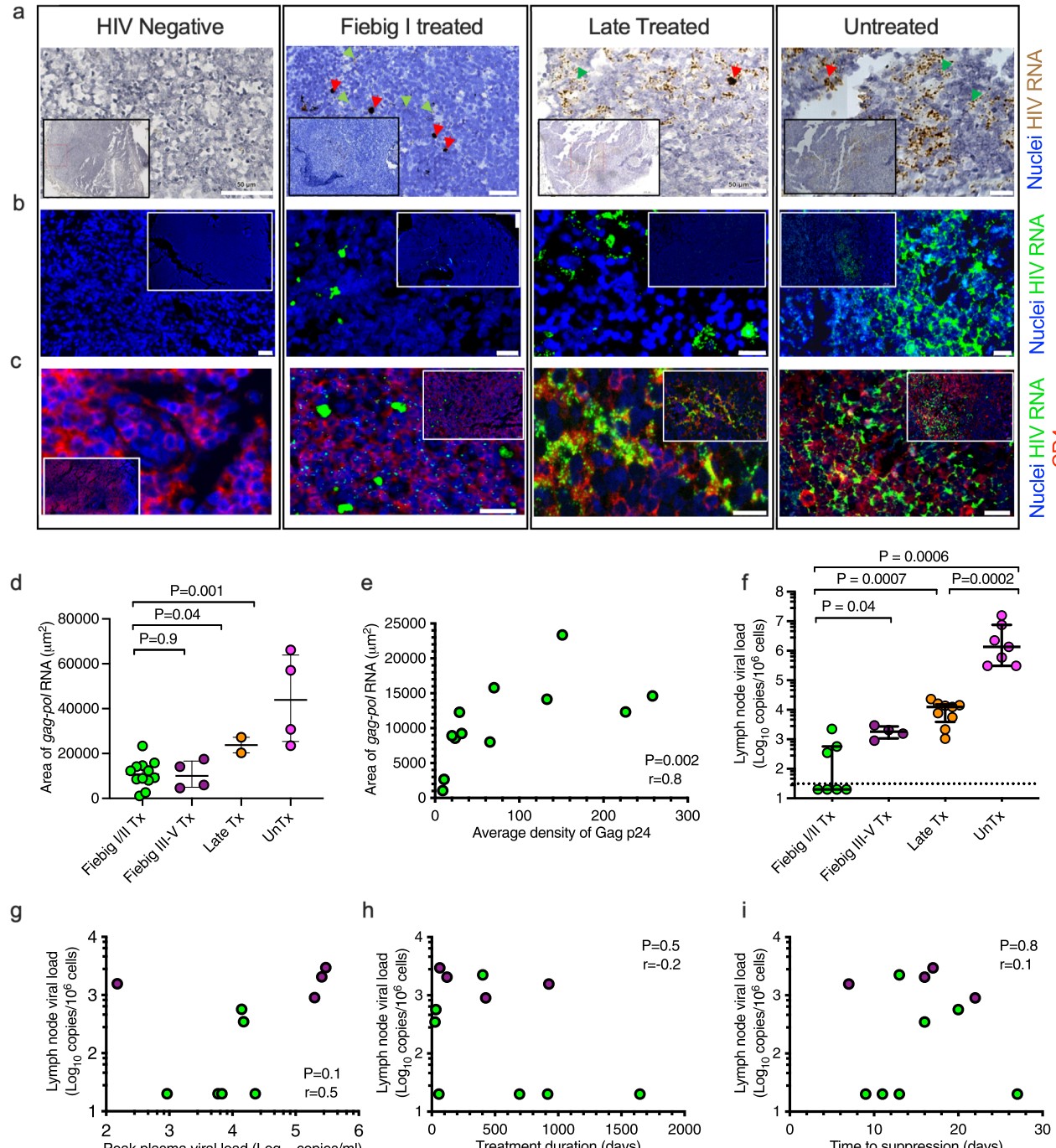

**Fig. 2 HIV-RNA persistence in lymph nodes of Fiebig I/II treated individuals.** HIV-RNA detection in lymph nodes (LN) of Fiebig I/II treated individuals using RNAscope (**a**–**e**) and Cobas® AmpliPrep HIV-1 test (**f**–**i**). **a** RNAscope hybridization for HIV *gag-pol* RNA was detected using 3, 3'-diaminobenzidene (DAB, brown) or **b**, **c** fluorescent Opal polymers (green). Representative images for HIV negative, Fiebig I/II treated (Tx), late Tx and untreated (unTx) HIV-infected LN sections are shown. Single RNA transcripts are seen as punctate dots; clusters of transcripts are also observed. Red arrowheads identify HIV RNA+ cells and green arrowheads identify virions on follicular dendritic cells. **c** Images showing multiplexed RNAscope *gag-pol* hybridization (green) coupled with IF staining for CD4+ cells (red). Three independent experiments were conducted with similar results. Scale bars are 50 μm (**a**) or 20 μm (**b**, **c**). **d** RNA signals quantified in micrographs using Fiji [(ImageJ software, Fiebig I/II Tx, $n = 12$; Fiebig III–V Tx, $n = 4$; late Tx, $n = 2$; and unTx, $n = 4$). Five fields of view are analyzed per sample and averaged. **e** A correlation analysis of area staining of *gag-pol* RNA and Gag p24 density for Fiebig I/II Tx LNs. **f** Viral RNA loads are quantified in lymph node mononuclear cells (LNMCs) (Fiebig I/II Tx, $n = 7$; Fiebig III–V Tx, $n = 4$; late Tx, $n = 9$; and unTx, $n = 7$). Viral loads below the limits of detection of the assay are assigned a value of 20. Correlation analysis of LNMCs' viral loads with **g** peak plasma viral loads, **h** treatment duration, and **i** time to suppression for Fiebig I/II Tx, $n = 7$; and Fiebig III–V, $n = 4$; donors. All statistical tests are two-sided and $p$ values are from the Mann–Whitney $U$ test (**d**, **f**). Spearman rho ($r$) values and $p$ values are reported for correlation analyses (**e**, **g**–**i**). Dotted line denotes threshold of viral load detection. Error bars represent interquartile range (**d**, **f**). Data are presented as median and interquartile range (**d**, **f**). Source data are provided as a source data file.

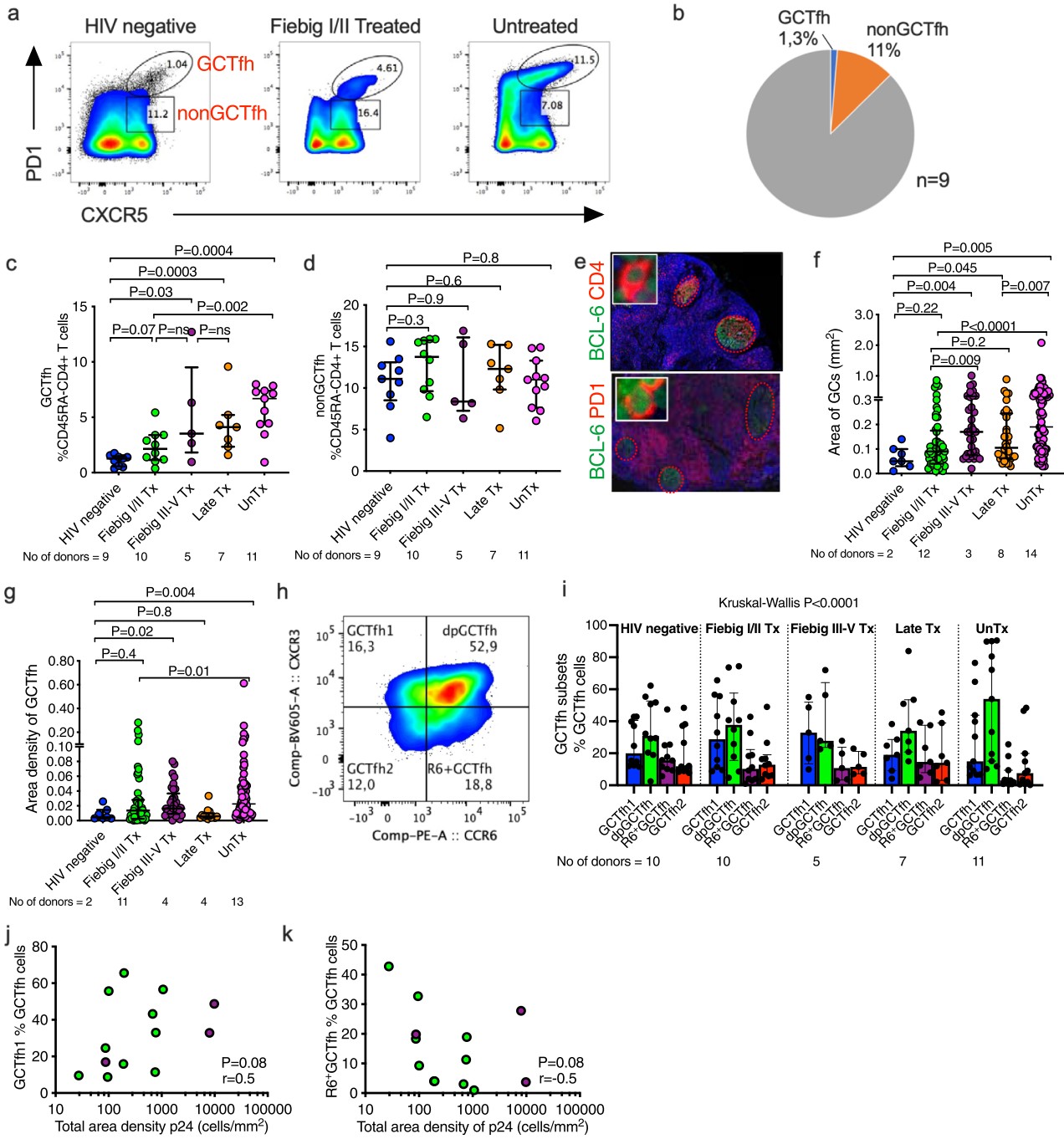

**Fig. 3 Expansion of Germinal center T follicular helper (GCTfh) cells during HIV-1 infection. a** Representative flow cytometry plots, **b** pie chart and **c** summary plots comparing the proportions of GCTfh (CXCR5hiPD-1hi), and **d** nonGCTfh (PD-1+CXCR5+) cells in HIV negative (HIVneg), Fiebig I/II treated (Tx), Fiebig III–V Tx, Late Tx and untreated HIV-infected (unTx) groups. **e** Representative images showing lymph node sections stained with BCL-6 (green) to define germinal centers and CD4 (red), or PD-1 (red) to localize GCTfh cells. Three independent experiments were conducted with similar results. Aggregate results for the (**f**) area of GCs and (**g**) area density of GCTfh cells computed using TissueQuest (TissueGnostics, Vienna) software. **h** Representative flow cytometry plot showing gating for Tfh subsets and **i** aggregate data across the groups. **j** Correlation analysis of Gag p24 density measured using image cytometry with GCTfh1 (CXCR3+CCR6−) and **k** R6+GCTfh (CXCR3−CCR6+) subsets distribution measured using flow cytometry (n = 13). All statistical tests are two-sided and p values are from the Mann–Whitney U test (**c**, **d**, **f**, **g**, **i**). Spearman rho (r) and p values are reported for correlation analysis (**j**, **k**). The number of donor samples analyzed in each group is indicated. Error bars represent interquartile range. Data are presented as median and interquartile range (**c**, **d**, **f**, **g**, **i**). Source data are provided as a source data file.

Lastly, given the notion that HIV-specific CD4+ T cells might be more susceptible to infection and contribute to viral persistence[39], we used class II tetramers to characterize HIV-specific Tfh responses in LN tissues. DRB1*11:01 and DRB1*13:01 class II tetramers previously described[38,40] were used to identify HIV-specific GCTfh and nonGCTfh cells in 3 Fiebig I/II Tx, 2 late Tx and 4 unTx donors within our cohort expressing the class II DRB1*11:01 and DRB1*13:01 alleles (Supplementary Fig. 3l, m). These results demonstrate that HIV-specific Tfh responses are induced during early Tx HIV infection.

However, we did not have sufficient tetramer+ data to determine if antigen specific CD4+ T cells are preferentially infected.

**CXCR3+ Tfh cells harbor a greater burden of persistent HIV RNA in lymph nodes obtained from treated individuals with sustained plasma viral suppression.** Since image analysis for Gag p24 indicated that most of the HIV antigen was confined within discrete regions of GCs, and flow data showed differential correlation between levels of Gag p24 and GCTfh subsets, we next stained for other markers shown to be highly expressed on human Tfh[38]. We also used FDC markers to identify residual Gag p24 that has been reported to persist on FDCs[35]. IF imaging of serial sections stained with different combinations of antibodies and detected with Opal fluorophores revealed that Gag p24 co-localized with several phenotypic markers (Fig. 4a and Supplementary Fig. 4), including PD1 (Fig. 4a, i), CD4 (ii), CXCR3 (iii, iv), CCR6 (iv) and FDC (v). To more definitively identify the Tfh subset that habored the most HIV infection burden, we quantified HIV RNA in LNMCs isolated from LN tissue of 3 Fiebig I/II Tx, 2 Fiebig III–V Tx and 3 late Tx donors and FACS-sorted into the 4 different Tfh subsets (Fig. 4b). HIV mRNA was detectable using digital droplet PCR in all the subsets (Fig. 4c). Importantly, when we analyzed the cells based on expression of chemokine receptors, CXCR3 and CCR6, we found that CXCR3+ Tfh subsets harbored significantly greater amounts of HIV RNA than other subsets ($p = 0.003$, Fig. 4c).

To further interrogate preferential infection of CXCR3+ Tfh cells, we used a broadly neutralizing antibody called 3BNC117 to stain HIV-infected cells expressing the HIV envelope protein (gp120), while simultaneously staining for CXCR3. 3BNC117 targets the CD4 binding site on the surface of HIV-1 Envelope (Env) glycoprotein[41]. Assay validation showed clear 3BNC117 staining of LNMCs that were infected with NL4-3 in vitro for 7 days compared to uninfected control (Fig. 4d). Further validation showed ex vivo staining of LNMCs of a viremic donor with no staining observed for two HIV negative donors (Fig. 4e). Having validated the assay, we performed ex vivo staining of seven paired LNMC and PBMC samples obtained from seven Fiebig I/II treated donors. Representative flow plots for one donor and aggregate data for seven donors showed detection of HIV-1 Env (3BNC117) positive LNMCs at significantly greater frequency compared to paired PBMC samples ($p = 0.03$, Fig. 4f). To confirm detection of low frequency HIV-1 positive cells ex vivo, we intracellularly stained aliquots of the same samples with anti-Gag p24 antibody. Similarly, Gag p24+CD4+ T cells were readily detectable in LNMCs compared to PBMCs ($p = 0.01$, Supplementary Fig. 5a). We phenotyped infected cells by dual staining of 3BNC117 and CXCR3 and observed a trend toward more Env+CD4+ T cells co-expressing CXCR3 ($p = 0.06$, Fig. 4g) than those not expressing CXCR3. Together, these data suggest that CD4+CXCR3+ expressing Tfh cells may be preferentially infected in vivo compared to other subsets.

**Impact of HIV-specific CD4+ and CD8+ T cell responses on HIV persistence in the lymph node during ART.** We previously showed that immediate ART initiation augments HIV-specific T cell function in PB[24]. To investigate the effects of early ART on LN responses, we begun by investigating if there were compartmental differences in the frequency of HIV-specific responses between LN and PB. We used intracellular cytokine staining (ICS) to measure the proportions of HIV-specific CD4+ and CD8+ T cells in LNs and paired blood samples using 14 fully suppressed Fiebig I/II Tx donors on uninterrupted therapy for greater than a year. Representative flow plots for one donor and aggregate data show significantly higher frequency of PB Gag-

specific CD8+ T cells ($p = 0.05$) compared to LN responses (Fig. 5a). HIV-specific CD4+ T cell frequencies also trended toward greater frequencies in PB relative to LN ($p = 0.06$; Fig. 5b). Next, we investigated whether HIV-specific CD8+ T cell responses limit HIV persistence in the LN, and found a negative correlation between the frequency of HIV-specific LN CD8+ T cell responses and HIV Gag p24 density ($p = 0.02$, $r = -0.7$; Fig. 5c). There was no correlation observed between peripheral CD8+ T cell responses and the amount of persistent Gag p24 antigen in the LN, suggesting the peripheral responses may not accurately depict HIV persistence in LNs. Notably, there was no correlation between LN or peripheral CD4+ T cell responses and persistent HIV Gag p24 in the LN (Fig. 5d).

Considering that proliferative CD8+ T cell responses are often associated with protection[42,43], we next measured virus-specific responses by carboxyfluorescein succinimidyl ester (CFSE) dilution. Representative flow plots for a donor with low and a donor with high Gag p24 density are shown (Fig. 5e). Aggregate data show proliferative Gag-specific CD8+ and CD4+ T cell responses negatively correlated with HIV Gag p24 burden (both CD8 and CD4, $p = 0.04$, $r = -0.7$; Fig. 5f). Together, these data show an association between maintenance of functional cellular responses and reduced HIV viral antigens in LNs.

Given most of the residual virus was concentrated within GCs, we next assessed the capacity of HIV-specific T cell responses to traffic into the GCs by enumerating the frequencies of CXCR5+ HIV-specific responses in LN, which denote capacity to migrate into GCs. Representative flow plot and aggregate data show lower CXCR5+ HIV-specific (gamma secreting) CD8+ T cells compared to CXCR5− CD8+ T cells among Fiebig I/II and III–V treated donors ($p = 0.001$), suggesting reduced capacity of CD8+ T cells to migrate into GCs (Fig. 5g). These data are consistent with our recent publication where we showed by IF imaging that CD8+ T cells are largely excluded from GCs[44]. These data partly explain the observed greater HIV antigen burden in GCs relative to extrafollicular areas in some donors. Additionally, there was no correlation between plasma CXCL-13 (part of the CXCR5-CXCL-19 axis crucial for recruitment of immune cells into GCs) and density of HIV antigens in LNs (Supplementary Fig. 5b). Together, these data show that reduced functional HIV-specific CD8+ T cell responses within GCs might contribute to HIV persistence in this tissue microenvironment.

**Discussion**

Comprehensive understanding of ART-mediated HIV suppression in tissue sanctuary sites is critical to the design, optimization, and evaluation of curative strategies. Moreover, a therapeutic vaccine for HIV-1 infection would need to induce robust anti-HIV immune responses in ART suppressed individuals to mediate post-treatment viral control. Here, we used a very well characterized cohort of persons with hyperacute HIV infection to conduct a comprehensive analysis of HIV persistence in LNs following ART initiation in Fiebig stage I/II and to elucidate Tfh cell responses which are critical for robust B cell and CD8+ T cell functions.

Most donors exhibited persistent HIV antigens in LN despite prompt blunting of initial peak viremia and sustained plasma viral suppression for as long as 55 months, suggesting that early therapy initiation may not fully eradicate persistent virus in lymphoid tissue sites. Immediate therapy reduced GCTfh expansion which is typically associated with dysregulation of B cell responses due to excessive GC reactions in unTx HIV infection[36,45]. Moreover, mitigated GCTfh responses decreased the number of cellular targets of HIV infection. Importantly, the association between functional

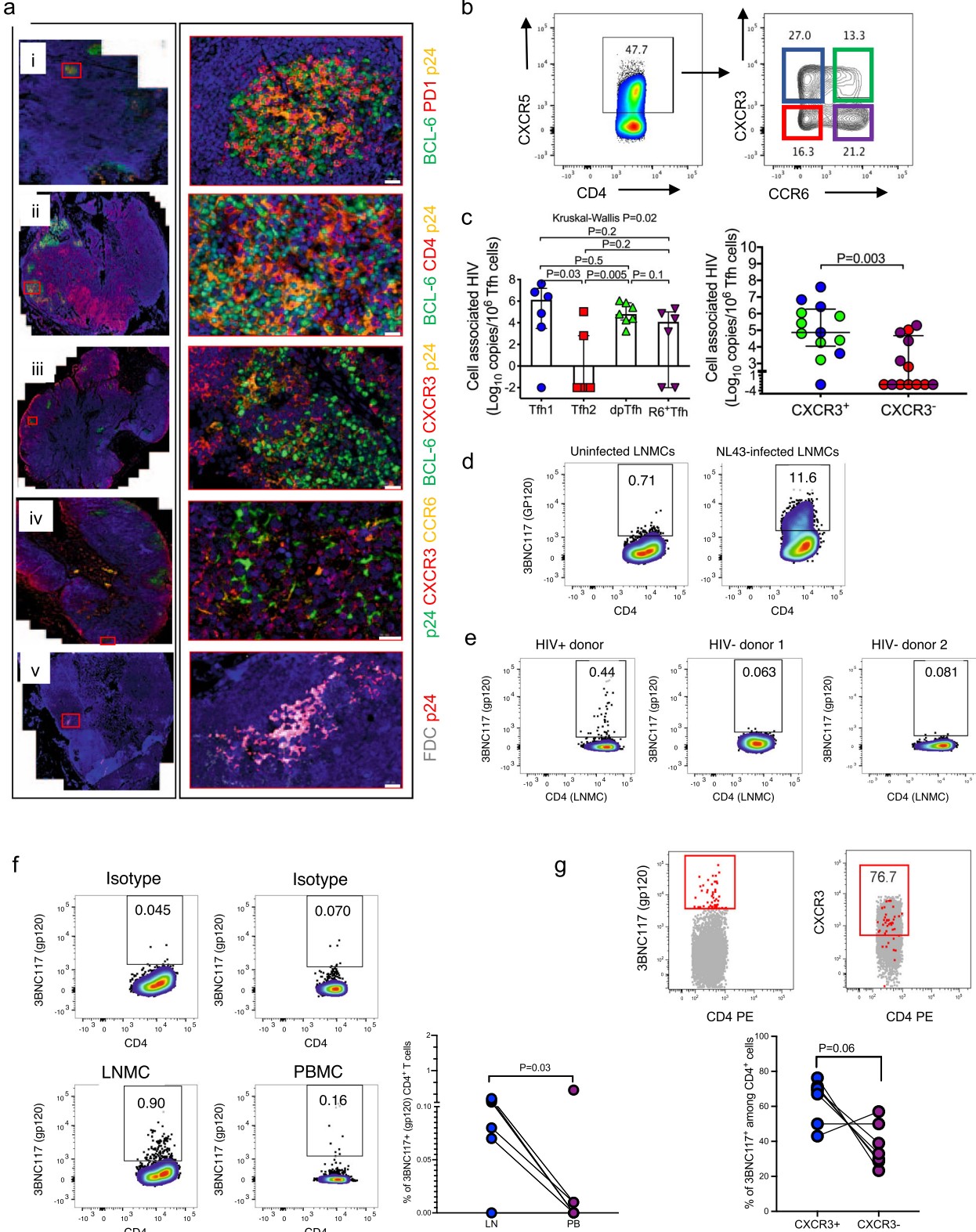

immune responses and reduced viral burden in LNs indicates that T cell responses contribute toward elimination of infected cells during therapy. Combined, these data highlight the need to prioritize elimination of active HIV persistence in LNs as a critical step to achieving a cure or prolonged HIV remission off therapy.

Our unique ability to obtain excisional LN biopsies in the FRESH cohort allowed for characterization of sites of virus persistence within the LN architecture in persons in whom peak viremia is blunted. The topological analysis of persistent HIV antigens within intact LN tissues identified greater HIV protein antigen burden within B cell follicles in some patients. Notably,

**Fig. 4 CXCR3$^+$ Tfh cells contribute to HIV persistence in treated hyperacute HIV-1 infection.** Representative IF images characterizing HIV Gag p24$^+$ cells in the germinal centers (BCL-6$^+$, green). **a** The co-localization of Gag p24 antigens (yellow color in (i–iii), green in (iv), red in (v)) with cells expressing (i) PD1$^+$ in red, (ii) CD4$^+$ in red, (iii) CXCR3$^+$ in red, and (iv) CCR6$^+$; yellow color, surface markers and (v) follicular dendritic cells (FDC, gray color) are assessed by immunofluorescence microscopy. Green, red, and yellow signals are from Opal fluorophores 520, 570 and 690 (PerkinElmer) and nuclei are counterstained with DAPI (blue). **b** Representative flow cytometry plots showing gating for FACs-sorted Tfh subsets. **c** HIV mRNA quantified in FACs-sorted Tfh subsets (from $n = 8$ donors) using digital droplet PCR. Absolute numbers of quantified HIV transcripts are equated to absolute cell numbers determined using the expression of β2M. Amounts of HIV mRNA within CXCR3$^+$ and CXCR3$^-$ subsets are also compared. **d** In vitro NL4-3 infected and uninfected LNMCs surface stained with 3BNC117 monoclonal antibody. **e** Flow plots of showing ex vivo 3BNC117 LNMC staining for one HIV positive and two HIV negative donors. **f** Representative flow plot and aggregate data show proportion of 3BNC117$^+$ CD4$^+$ T cells in LNMC and paired PBMCs for seven donors. **g** Representative flow plot and aggregate data show proportion of 3BNC117$^+$ CD4$^+$ T cells that either co-express or do not express CXCR3. Statistical differences are calculated using Mann–Whitney $U$ (**c**, **f**, **g**) and Kruskal–Wallis (**c**) tests and all statistical tests are two-sided. Error bars represent interquartile range (**c**). Data are presented as median and interquartile range (**c**). Source data are provided as a source data file.

onset and duration on therapy did not significantly affect the amount of detectable Gag p24 protein, consistent with the notion of rapid HIV reservoir establishment followed by very slow decay rate[46]. Moreover, there are variable decay dynamics between active and latent HIV reservoirs. Active HIV reservoirs which are majorly responsible for low-level viremia during ART have been implicated in higher virological failure rates, persistent immune activation and inflammation[47]. Thus, the need for identifying all sources of persistent virus during ART. Importantly, our data suggest that LN GCs may be major sites of HIV persistence, with the potential to be a source of rebound viremia upon treatment interruption.

Using a highly specific ISH assay, RNAscope[48,49], which, in addition to HIV antigen detection provided further evidence in support of persistent HIV transcription in LNs in the face of ART in 86% of very early treated donors. We identified densely spherical signals by RNAscope staining in some early treated individuals suggestive of productively infected viral RNA$^+$ cells, consistent with a previous report in which active HIV RNA transcription was detected in the LNs of patients who initiated therapy in the chronic phase of illness[13]. Moreover, our data reveal heterogeneity in the amount of persistent HIV transcripts in very early treated aviremic individuals despite similar levels of peak viremia and rapid plasma viral suppression kinetics following ART initiation. Viral antigen persistence and ongoing transcription could indicate that HIV continues to cause immune damage in anatomical sites despite full suppression in PB, but it may also suggest that even in early treated individuals, priming, stimulation and harnessing of HIV-specific immunity for curative strategies will not be insurmountable because functional HIV-specific immunity is preserved.

Identifying cellular reservoirs of HIV in tissues has been a major area of research (reviewed in ref. [50]). These studies describe GCTfh subset compositions anatomically and phenotypically during HIV infection and their contributions to persistent virus. The observed positive correlation between the proportions of GCTfh1 cells (which are CXCR3$^+$CCR6$^-$GCTfh) and detection of greater amounts of HIV RNA relative to CXCR3$^+$ Tfh cells, indicates that CXCR3 may yet be another phenotypic marker of Tfh cells that have greater HIV transcription activity on ART. These data are consistent with a study that reported greater amounts of SIV DNA in CXCR3$^+$ GCTfh compared to CXCR3$^-$ GCTfh in macaques[51]. It is reasonable to attribute increased HIV burden in the CXCR3$^+$ Tfh subset to CXCR3 being used as an alternative co-receptor for HIV entry, which has previously been reported[52]. However, confirmatory work is needed. In any case, we have identified a marker for HIV-infected cells during ART that could be targeted for elimination as part of an HIV eradication strategy. Whether or not the CXCR3$^+$ Tfh population identified in this study represents the same population as the PD-1$^+$ subset that was recently implicated in HIV persistence is intriguing and warrants further investigation[13,53].

Low CD8$^+$ T cell density in GCs is thought to be a major reason for persistently high HIV antigen burden in this anatomical niche[54,55]. Interestingly, we detected significantly greater proliferative responses in individuals with little to no detectable HIV antigens compared to those with greater LN HIV antigen burden. It is difficult to determine if the low antigen environment leads to the development of better CD8$^+$ T cell function or whether superior CD8$^+$ T cell functionality results in lower antigen burden in LN tissue. However, we did not have longitudinal data to unpack this conundrum. Nevertheless, these findings are consistent with our recent study showing that very early ART is associated with functionally superior cellular responses[24]. Indeed, other studies have demonstrated the cytolytic activity of CXCR5$^+$ CD8$^+$ T cells against LCMV and HIV-infected cells[56,57]. Together, these data suggest that HIV-specific T cell responses contribute to HIV suppression in LN during therapy[58]. However, longitudinal studies using serial biopsies from the same donor are needed to confirm these findings.

A notable limitation of the study is that we could only obtain one LN sample per study participant, thus we could not conduct intra-individual longitudinal HIV decay kinetics, or longitudinally define the immune responses in tissues associated with control. This limitation was partly overcome by sampling LNs from many participants over a very wide time range. Serial LN studies have been reported in the absence of complications[59]. Alternatively, future studies could attempt serial fine needle aspirates (FNA), which is a quick procedure, less invasive and more amendable to serial sampling[60]. This approach results in loss of spatial information. Moreover, cell yields from FNAs may be limiting for some of the studies described here, nevertheless, cell subset distribution in FNAs is representative of whole LNs and will be advantageous for the characterization of tissue cell subsets, HIV reservoir quantification among other LN tissue studies[60]. Additional limitations of our study are the lack of gender and age diversity within the study population. Although we studied mainly young women, this is the group at disproportionate risk of infection and women are underrepresented in most studies to date[61]. Higher immune activation[62] among other sex-linked immunological factors in HIV infection may prevent the generalization of our results to HIV-infected males. In addition, well described structural changes to the LN with age including, collagen deposition and fibrosis[63,64], might dampen immune responses during early ART. These issues should be addressed in future studies. As with many immunology studies, sample sizes and quantities are a limiting factor. Specifically, due to the subtle differences between the early Fiebig stages, much larger samples sizes in the groups might be useful to tease out subtle but clinically relevant differences.

In conclusion, our results demonstrate HIV persistence in LNs despite prompt and durable ART-mediated plasma viral suppression. HIV structural proteins and HIV RNA persist in LNs of in 12 of 14 individuals, albeit at lower levels compared to

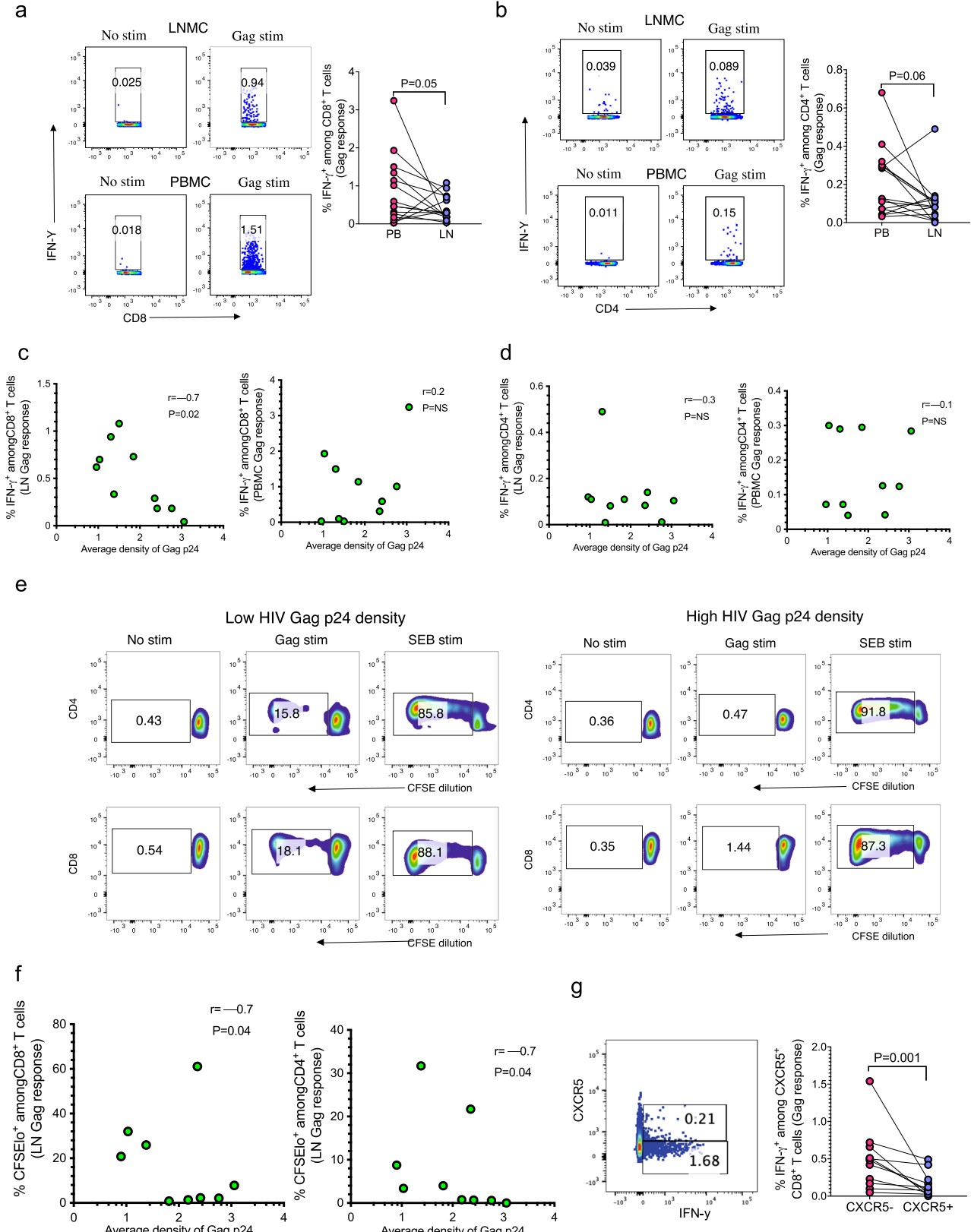

treatment in chronic infection. Given sample limitations, one cannot rule out persistent infection in the other two individuals. Among those with detectable infection, GCs serve as the primary anatomical site of HIV persistence and the major cellular source is CXCR3 expressing Tfh cells. Together, our results emphasize

the importance of very early initiation of ART in Fiebig stage I/II to reduce the amounts of persistent virus in the LN and highlight the need for interventions to completely eradicate residual viremia in immune privileged and anatomically compartmentalized tissue sites.

**Fig. 5 HIV-specific CD8$^+$ T cell responses limit the amount of persistent HIV antigens in lymph nodes during ART. a, b** Intracellular cytokine staining was conducted after stimulating PBMCs and lymph node mononuclear cells (LNMCs) with HIV-Gag. **a** Representative flow cytometry plots and aggregate data of 14 donors showing IFN-γ$^+$ CD8$^+$ T cells, and **b** IFN-γ$^+$ CD4$^+$ T cells after stimulation with HIV-1 clade C Gag peptide pools. Correlation analysis of average Gag p24 density; measured from image analysis, with the frequency of **c** IFN-γ$^+$CD8$^+$ T cells and **d** IFN-γ$^+$CD4$^+$ T cells in LNMCs and PBMCs. **e** Representative flow cytometry plots of CFSE-labeled CD4$^+$ and CD8$^+$ T cells after 7-days of stimulation of LNMCs with HIV-1 clade C Gag peptide pools. **f** Aggregate data correlating CFSEloCD8$^+$ and CFSEloCD4$^+$ T cell responses and Gag p24 density. **g** Representative flow plot and aggregate data showing frequency of HIV Gag-specific CXCR5$^+$ CD8$^+$ T cells. All statistical tests are two-sided and p values are from the Mann–Whitney U test (**a, b, g**). Spearman rho (r) and p values are reported for correlation analysis (**c, d, f**). Source data are provided as a source data file.

## Methods

**Study approval**. All study participants provided written informed consent prior to inclusion in the study. Ethical approval for the study was granted by the University of KwaZulu-Natal Biomedical Research Ethics Committee (protocol number BF298/14) and the Institutional Review Board of Massachusetts General Hospital (protocol number 2015-P001018).

**Study population, samples, and performance site**. Study participants were drawn from the HIV Pathogenesis Programme (HPP) lymph node study (LNS) cohort, Durban, South Africa. Recruitment into the HPP LNS cohort were from the FRESH cohort described in ref. [16] (n = 41) and another primary HIV infection cohort in Durban, South Africa where participant's time of infection is less defined (n = 35). Axillary, cervical or inguinal LNs were surgically excised at Prince Mshiyeni Hospital in Umlazi, and 120 ml paired PB was also obtained from each participant. Viral load measurements were performed by HIV-1 RNA testing using the NucliSens EasyQ v2.0 assay (BioMérieux Clinical Diagnostics, Marcy-l'Étoile, France), through a certified commercial laboratory. CD4$^+$ T cell counts were enumerated by Tru-Count technology and analyzed on a FACSCalibur flow cytometer (Becton Dickinson (BD) New Jersey, USA). Sample processing and laboratory studies were performed at the Africa Health Research Institute in Durban, South Africa.

**Lymph node and blood sample processing**. Excised LNs were sectioned into two, one section was fixed in 10% formal-saline (Sigma-Aldrich, St. Louis, Missouri, USA) for IF microscopy studies, while the second section was macerated to release LNMCs according to the method of Schacker et al.[65]. The cells were passed through a mesh screen and harvested by centrifugation (625 × g, 6 min, room temperature (RT)].

Peripheral blood mononuclear cells (PBMCs) were isolated from patient's blood samples by density-gradient centrifugation using Histopaque-1077 (Sigma-Aldrich) and cryopreserved in liquid nitrogen[66].

**Viral RNA quantification in lymph node mononuclear cells (LNMCs)**. Cryo-preserved LNMCs (10 million cells) were lysed, and viral RNA was quantified using the Cobas® AmpliPrep HIV-1 test (Roche, Mannheim, Germany) at an accredited clinical laboratory using standardized protocols.

**Immunofluorescence (IF) microscopy**. IF microscopy staining was performed on 4 μM sections of formalin-fixed paraffin-embedded (FFPE) LNs using the Opal 4-color fluorescent IHC kit (PerkinElmer, Waltham, MA, USA). Sections were deparaffinized using xylene (Honeywell research chemicals) and rehydrated, prior to antigen retrieval using AR6 buffer (20 min, 100 °C, (PerkinElmer)). Next, two blocking steps (2 × 10 min, RT) were performed with the Dako peroxidase-blocking reagent (Agilent Technologies, Glostrup, Denmark) and Bloxall block (Vector Laboratories, Burlingame, CA, USA). The slides were washed with 0.05% Tween 20 in Tris-buffered saline (TBS-T) for 5 min, sequentially probed with the primary antibody (30 min, RT), and Opal polymer HRP (20 min, RT (PerkinElmer)) and detected using the Opal polymer 520 (10 min, RT). This protocol was repeated for the second and third antibodies with Opal polymers 570 and 690 respectively, followed by counterstaining with spectral DAPI (PerkinElmer) to make a total of four different fluorochromes. Primary antibodies used in these combinations include anti-human BCL-6 ((clone PG-B6p) Dako/Agilent Technologies), CCR6 ((R6H1) Thermo Fisher Scientific, Waltham MA, USA), CD4 ((clone 4B12) Dako/Agilent Technologies), CXCR3 ((clone 6H1L8) Thermo Fisher Scientific), FDC ((cloneCNA.42), Dako/Agilent Technologies) p24 ((clone Kal-1), Dako/Agilent Technologies), and PD-1 (clone NAT105) Abcam, Cambridge, MA, USA). Slides were mounted with Dako fluorescence mounting medium (Agilent Technologies) and imaged with the Axio Observer, ×20 objective lenses, a Hamamatsu C13440-20C camera and TissueFAXS imaging software (TissueGnostics, Vienna, Austria).

**RNAscope® in situ hybridization (ISH)**. RNAscope® ISH was conducted using the RNAscope® 2.5 HD assay kit (Advanced Cell Diagnostics (ACD), Newark, CA, USA, Cat No: 322300) and the RNAscope® multiplex fluorescent kit v2.0 (ACD, Cat No: 323100) as per manufacturer's instructions. Briefly, pre-treated samples were hybridized with the clade C HIV-1 gag-pol probe (Cat No: 317691) at 40 °C

for 16 h. Next, the samples were incubated with signal amplification probes and horseradish peroxidase conjugated secondary antibodies. The signal was detected with either diaminobenzidine for the RNAscope® 2.5 HD assay (ACD) or with Opal fluorophores (PerkinElmer) for the multiplex fluorescent assay. Slides were imaged with Axio Observer and TissueFAXS imaging software (TissueGnostics).

**Quantitative image analysis**. Quantitative image analysis of Gag p24 in IF images of whole tissue section scans was conducted with TissueQuest software (TissueGnostics). Two independent experiments of total area measurements and nuclear segmentation analyses were performed on each whole tissue scan. The numerical data generated from the analyses are displayed in scattergrams. Grey-scale images were analyzed and each channel was processed separately by the software using DAPI as a master marker. In cases where images were stained with another nuclear marker such as BCL-6, then the FITC channel was used as a virtual channel for nuclei identification. Negative control slides were used to set the threshold values in the scattergrams and to distinguish specific staining signals from non-specific or background fluorescence signals. Although HIV Gag p24 staining was generally intense, there was no notable spillover of the signal to other channels (Supplementary Fig. 1a, b). Also, p24 co-staining was only observed with FDCs and CD4 markers but not CD8 cells (Supplementary Fig. 1c, d).

Analysis of gag-pol RNA signals was done using Fiji, an open-source software based on ImageJ (ImageJ version: 2.0.0-rc-69/1.52p) which is optimized for biological image analysis[29]. Briefly, images were segmented using the color segmentation plugin with the algorithm for Hidden Markov Model. Thresholding was applied to the segmented image and the total area of brown RNA signals was measured and recorded. Five images were analyzed per sample and averaged. Pixel measurements were converted to μm using the scale bar.

**Flow cytometry analysis**. Freshly isolated or frozen LNMCs and PBMCs were characterized using flow cytometry analysis with standardized protocols[67]. Cells were stained with LIVE/DEAD Fixable Blue dead cell stain kit (Thermo Fisher Scientific), CD3 Brilliant Violet (BV) 711 (BioLegend, San Diego, CA, USA), CD8 BV786 (BD Biosciences, San Jose, CA), CD4 BV650 (BD Biosciences) CXCR5 Alexa Fluor (AF) 488 (BD Biosciences), PD-1 BV421 (BioLegend), CCR6 Phycoerythrin ((PE) BioLegend), CXCR3 BV605 (BioLegend) and CD45RA PE-Cyanine (Cy)−7 (BioLegend), for 30 min at RT.

For ICS, PBMCs or LNMCs were either left unstimulated or stimulated with HIV clade C overlapping peptide (OLP) pools spanning Gag, Nef, or Env proteins or Staphylococcal enterotoxin B (SEB, 0.5 μg/ml) in the presence of GolgiStop and GolgiPlug protein transport inhibitors (BD Biosciences) for 16 h at 37 °C, prior to surface staining with the panel of antibodies comprising LIVE/DEAD fixable Aqua dead cell stain (Thermo Fisher Scientific), CD3 BV711, CD4 BV650 and CD8 BV786. After fixation and permeabilization with the BD Cytofix/Cytoperm kit (BD Biosciences), cells were again stained using TNF-α A700 (BD Biosciences) and IFN-γ PE-Cy7 (BioLegend) antibodies.

T cell proliferation was measured by labeling LNMCs with CFSE, stimulating cells with HIV clade C OLP pools for 7 days and staining with CD3 BV711, CD4 BV650 and CD8 BV786. Stained cells were acquired using an LSRFortessa (BD Biosciences) with FACSDiva™ software or sorted using the FACS Aria Fusion (BD Biosciences). Data were analyzed using the FlowJo version 10.0.8 (Flowjo, LLC, Ashland, Oregon).

HIV-infected cells were identified by surface staining with biotinylated 3BNC117 antibody followed by streptavidin PE (Thermo Fisher Scientific) and/or intracellular staining with HIV Gag p24 RD1 ((clone KC57), Beckman Coulter, Indianapolis, USA) after fixation and permeabilization with Cytofix/Cytoperm (BD Biosciences).

**HLA class II tetramer studies**. HIV-specific Tfh responses were defined using fluorochrome conjugated HLA class II tetramers. Briefly, cells were stained for 1 h at 37 °C with APC and PE conjugated HLA Class II tetramer complexes, washed in 2% FCS-PBS and then stained with these antibodies: LIVE/DEAD Fixable Blue dead cell stain kit (Thermo Fisher Scientific), CD3 BV711 (Biolegend), CD4 BV650 (BD Biosciences), CD8 BV786 (BD Biosciences), CXCR5 AF488 (BD Biosciences), CXCR3 BV605 (Biolegend), PD-1 BV421 (Biolegend) and CD45RA AF700 (Biolegend); for 20 min at RT. Cells were washed and acquired on the LSRFortessa (BD Biosciences).

**Digital droplet PCR**. Total RNA was extracted from FACS-sorted LNMC Tfh subsets using Qiagen RNeasy kit (Qiagen) after lysing cells with QIAzol lysis reagent (Qiagen, Hilden, Germany) according to manufacturer's instructions, and used for cDNA synthesis using the iScript cDNA synthesis kit (Bio-Rad, Hercules, CA, USA). The cDNA was used as a template for HIV *gag* mRNA quantification by TaqMan digital droplet PCR assay using custom probes (Assay ID: APCE4R6, Thermo Fisher Scientific) in a two-step digital droplet PCR reaction. PCR thermal cycling was conducted following optimized cycling conditions: an initial denaturation at 95 °C for 10 min, 40 cycles of 30 s at 94 °C, 1 min at 60 °C, followed by a final incubation at 98 °C for 10 min and holding at 4 °C until reading time. After PCR amplification, droplets were measured in the QX200 ddPCR Droplet Reader (Bio-Rad), and target gene copy number was analyzed using QuantaSoft analysis software (Bio-Rad) and recorded as mRNA copies/20 µl. Absolute *gag* mRNA counts were normalized to the expression of the housekeeping gene β2M.

**Statistical analyses**. All statistical analyses were conducted with GraphPad Prism version 7.0 for macOs (GraphPad Software, San Diego, California, USA) and p values were considered significant if <0.05. Specifically, the Mann–Whitney $U$ and Kruskal–Wallis $H$ tests were used for group comparisons. Additional post hoc analyses were performed using the Dunn's multiple comparisons test. Correlations between variables were defined by the Spearman's rank correlation test.

**Reporting summary**. Further information on research design is available in the Nature Research Reporting Summary linked to this article.

## Data availability

Source data are provided with this paper.

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

## Acknowledgements

We would like to thank our study participants, the laboratory and clinic staff at FRESH and the HIV Pathogenesis Programme (HPP), Durban, South Africa and Drs Shiv Pillai and Takashi Maehara of the Ragon Institute of Massachusetts General Hospital, Massachusetts Institute of Technology, and Harvard University, Cambridge, MA, USA, for training on immunofluorescence microscopy. We acknowledge Dr. Faatima Laher of HPP, Drs. Jackson Marakalala and Daniel Muema of the Africa Health Research Institute, Durban, South Africa for technical assistance with digital droplet PCR, CXCL-13 ELISA and soluble CD14 ELISA studies, respectively. We would like to acknowledge the following funding sources; HHMI International research scholar award (Grant #55008743 to Z.M.N.); the US National Institutes of Health (NIAID) (R01AI145305 to Z.M.N., R37 AI67073 to B.D.W.); a Dan and Marjorie Sullivan Research scholar 97 Award (Grant # 224910 to Z.M.N.); and a Doctoral Innovation Scholarship from the South African National Research Foundation (NRF) (2014-2016 to O.O.B.) and the South African Research Chairs Initiative to T.N. Additional funding awarded to Z.M.N. was from the Mark and Lisa Schwartz Foundation, the Bill and Melinda Gates Foundation, the International AIDS Vaccine Initiative (IAVI, grant number UKZNRSA1001), the Ursula Brunner Fund at the Ragon Institute and the Victor Daitz Foundation. This work was also partially supported by Gilead Sciences Incorporated (to T.N.) and the Sub-Saharan African Network for TB/HIV Research Excellence (SANTHE), a DELTAS Africa Initiative (grant # DEL-15-006 to T.N.). The DELTAS Africa Initiative is an independent funding scheme of the African Academy of Sciences (AAS)'s Alliance for Accelerating Excellence in Science in Africa (AESA) and supported by the New Partnership for Africa's Development Planning and Coordinating Agency (NEPAD Agency) with funding from the Wellcome Trust (grant # 107752/Z/15/Z) and the UK government. The views expressed in this publication are those of the author(s) and not necessarily those of AAS, NEPAD Agency, Wellcome Trust or the UK government.

## Author contributions

Z.M.N., B.D.W., K.L.D. and T.N. initiated the study cohorts. Z.M.N. conceived the study. Z.M.N. and O.O.B. designed the experiments. ThaNg recruited the study participants, I.J. and J.P. performed the lymph node biopsies. N.I., ThaNk and A.M. processed the samples. O.O.B. performed all the experiments with the assistance of T.K., ThaNk, F.O., A.M., and C.C. under the supervision of Z.M.N. O.O.B. and Z.M.N. analyzed the data. O.O.B., Z.M.N. and J.M. wrote the manuscript. T.N. and B.D.W. edited the manuscript.

## Competing interests

The authors declare no competing interests.
