## [Peer Review File · Nature Communications]

CD8 lymphocytes mitigate HIV-1 persistence in lymph node follicular helper T cells during hyperacute-treated infectionREVIEWER COMMENTS

Reviewer #1 (Remarks to the Author):

The current manuscript addresses a very important issue related to the establishment of HIV tissue reservoir. The author applied complementary methods to address the viral dynamics in a prestigious cohort of relevant lymph nodes. The presented data support their claims. Interestingly, the data show that relevant blood dynamics do not reflect the tissue ones, further pointing to the need for future studies focusing on lymph node immunology/virology for the understanding of viral reservoir maintenance. Despite the merit of the presented data, several points need clarification, improvement in order to strength the author's claims.

Specific comments:

1. This reviewer appreciates the importance of the human material used in this study as well as the difficulties for access to lymph node biopsies from relevant tissues. The authors propose that future studies using FNA preparations could be useful at this point. The authors should comment on the advantages and disadvantages of such approach. The presented cohort is biased regarding the age of the participants. Given the additional lymph node abnormalities found in elderly, the authors should comment on the possible effect of such abnormalities in aging HIV infected subjects.
2. For a better appreciation of the tissue microenvironment on the observed dynamics, a general description (number of active follicles across the groups of donors, size of the active follicles, whether p24 or HIV RNA staining is associated with enlarged, lysed follicles or 'intact/preserved' follicular structures) should be provided.
3. Previous data have shown a reducing size of the lymph node Tfh cell latent reservoir with respect to time after cART initiation. The authors report no association between Gag p24 staining and treatment duration. Is this the case for the actively transcribed virus CD4 T cells (RNAscope)? If this is the case, the authors should comment on the possible dissociation between latent and 'active' reservoir of the virus.
4. Figure 1E: the data clearly show two groups of follicles even in the Fiebig I/II donors. Are the high p24 follicles originate from one or several donors? Is this profile associated with any of the other parameters under investigation?

5. Figure 3C: no difference was found between Fiebig I/II and late Tx donors. Similar profile was found when GC Tfh were calculated from generated images. Most likely this profile reflects the elimination of accumulated Tfh cells upon cART treatment. Still, it would be informative if the authors provide any information related to inflammatory cell types and abnormalities (e.g. fibrosis) in the tissues under investigation.

6. Figure 3F and text: the authors use the term GC Tfh 17, a term that was first introduced for 'circulating' Tfh cells. However, GC Tfh cells are not capable of producing IL17 even after PMA stimulation. The authors should use this terminology with caution.

7. Figure 3: it is known that several CD4 T cells within the follicular areas do not express Tfh-phenotypic markers. What are the numbers of total CD4 T cells in these follicles? What is the ratios of GC Tfh/total follicular CD4 T cells in these follicles?

8. Figure 4D and E: the authors should fix the gating for the intensification of specific cells (the current gates are too low and possibly contaminate the analysis outcome). Although the reviewer believes that the final profile will remain the same, the 'new' data will be more robust.

9. Lines 281, 283, 307: the text does not reflect what the corresponding figures show for the relevant populations (possibly a typo?)

Reviewer #2 (Remarks to the Author):

In this interesting tissue-focused analysis of HIV persistence and T cell immune responses in early and later-treated infection the authors demonstrate that HIV proteins and viral transcriptional activity persist for long periods of time in lymph nodes, even in those who are started during very early stages of infection (e.g. Fiebig I/II) and that presence of CD8 T cells in LN GCs was associated with lower measures of infectious burden.

Overall, this study shows that HIV persists in tissues over time despite early initiation of HIV therapy. This finding, in of itself, is not particularly novel but has several potential strengths given the nature of the cohort. For example, the cohort is comprised of women, who are vastly understudied in HIV persistence studies and makes this analysis refreshing. In addition, the study had access to excision

lymph node biopsies, allowing in situ hybridization characterization of viral proteins, RNA and infiltrating CD8 T cells in those treated earlier and later in infection.

The continued presence of HIV p24 in early treated LN is not surprising given that it is now understood that tissue seeding of HIV reservoirs happens quite early and likely persists indefinitely. The authors also mention that there were two early treated donors that had no detectable Gag p24, which is not surprising given only a single LN was excised and this should be acknowledged further in the discussion/limitations.

Figure 1 shows HIV Gag p24 in GCs, but 1F shows that HIV Gag can still be found up to 5-7% of the extra-follicular areas with the differences in GCs really driven by 4 participant samples (whereas the p24 is similar between EF and GC in the other participant samples). While this was statistically significant, it does appear that in a majority of participants, HIV gag was evenly distributed within and without GCs. Representative images of a patient with similar levels (which is the majority) should also be shown in panel D. As of now, it is possible that these figures were selected from the outliers as in panel F. These points should also be addressed more specifically in the results and limitations.

The analysis of immediate versus delayed therapy in terms of GCTfh expansion is interesting, but the way the data are presented are misleading. More specifically, the authors show that there is only a trend towards significance between Fiebig I/II and uninfected controls in terms of cell expansion versus slightly later initiation of ART led to significant differences in expansion between HIV infected and uninfected controls. However, this data does not necessarily support a direct comparison between early and delayed ART initiation and these data are not provided. In order to make the statement that there are differences between HIV treatment groups, a direct comparison is needed (I am assuming it was negative or would have presented in the paper). This data should be shown and indirect comparisons with two groups, and if there is not difference, should be highlighted in the paper.

The tetramer studies to identify HIV-specific CD4 populations is interesting but given only a handful of participant samples were analyzed, it seems difficult to draw a conclusion that the tetramer-specific GCTfh and nonGCTfh were at similar frequencies between untreated and Fiebig I/II, although the authors do acknowledge that there were insufficient quantities of tetramer + cells to determine if antigen specific CD4 T cells were preferentially infected.

One of the more interesting aspects of this study which seems to be downplayed a bit is the identification of cells that express surface gp120 in tissues in both early and later treated individuals. Whereas most of tissue-based analyses have looked at p24, there is little understanding of HIV env expression in the setting of treated HIV infection. This result is important as it suggests that various immunological modalities (such as CAR-T and bnAbs) may play an important role in reservoir eradication strategies. Figure 4 D and E however use isotype controls for flow staining on LNMC versus PBMC but a

more appropriate comparison would be uninfected LNMC and PBMC with 3BNC117. The reason for this is that the low numbers of positive cells both in PBMC and LN could be from non-specific binding of the bNab outside of HIV infection and negative tissue stained with 3BNC is really needed to determine if these cells are expressing HIV gp120.

I am not sure if SEB stimulation is the appropriate readout for deficiencies of CXCR5+ CD8+ T cells to secrete cytokines. It seems that differences in response to HIV peptide stimulations are far more relevant to the question at hand. Furthermore, prior work has already clearly demonstrated dichotomies between cytotoxicity of CD8 T cells within LN follicles and cytolytic potential/killing of cells in these compartments. At the least, measures of GzB/Perforin/CD107a etc. could provide a bit more information on HIV-specific responses rather than superantigen response to IFN production alone.

Fig 5 E is quite interesting but I think there may be a chicken/egg conundrum here. The data show that cells from low gag density have much more profound proliferation in response to various HIV gag/nef stimulation. As read, this could be simply that CD8 T cell responses with better HIV-specific proliferative capacity leads to reduced reservoir size. Could there also be dysfunctional responses caused by high HIV burden within these tissue regions? In other words, could a lower initial burden of HIV in tissue actually lead to a better, more efficient anti-HIV CTL response than those with much higher burdens of infection?

Of course, longitudinal data would certainly be able to clarify some of the above issues in more detail, but the authors make the valid limitation statement that excisional biopsies are morbid and can't be performed regularly on human participants.

The authors state that there is no correlation between circulating and LN HIV RNA, but sample size is somewhat limited and this seems a strong conclusion to make given the relatively small number of patients and heterogeneity in these HIV persistence measures.

Minor comments:

Overall, figure text is rather small on many figures and very difficult to read without amplifying the data plots on the screen. Larger font sizes would dramatically improve readability.

Should Mann U-Whitney be Mann Whitney U?

Could any of the findings observed in this study be secondary to female sex? Perhaps a bit more in discussion would be good.

unTx is not a standard abbreviation in manuscript text

Reviewer #3 (Remarks to the Author):

In this work, Ndhlovu et al performed an analysis of HIV-1 persistence in lymph node T follicular helper cells during hyperacute-treated HIV-1 infection. The group makes use of an unique cohort available in South Africa. The work is of high quality. A comparison of HIV persistence and HIV-specific T cell responses was done in LN biopsies obtained from patients that started treatment during different Fiebig stages. The work illustrates the importance of virus specific responses in sanctuaries sites.

A general question concerning the FRESH cohort: FRESH is a prospective study of uninfected 18-23-year-old women at high risk of HIV infection established at the epicenter of the HIV epidemic in South Africa, where yearly incidence rates approach 10%. Despite vigorous prevention efforts, twice weekly monitoring for viral RNA has identified and treated (Tx) persons at the onset of plasma viremia, allowing for immediate institution of ART in many cases resulting in peak plasma viral loads that are sometimes <1,000 RNA copies/ml and the preservation of CD4+ T cell numbers.

If this woman are at such a high risk, why aren't they on PREP? Is PREP offered to this woman?

Concerning persistence of GAG p24 antigen in GC: Quantitative image analysis of all treated LNs revealed greater area percent of Gag p24 staining in GCs compared to extrafollicular areas of the tissue ($P=0.04$, Fig. 1F). Together, these data demonstrate that early ART initiation in Fiebig stage I/II limits the magnitude of HIV Gag p24 antigen in LNs, but that Gag p24 can persist predominantly in follicular areas even after 4.5 years of fully suppressive treatment.

The data seems to be strong for Fiebig I/II versus the other Fiebig groups. The comparison GC versus EF is much weaker with a $p=0.04$. The data is drive by 4 patient/LN only. So, I do not agree one can make that claim. In addition, I was wondering whether this is only for Fiebig I/II patients, or is this claim still standing when taking into account all fiebig stages.

Concerning RNA in LN:

Figure 2D: why is the GAG-POL RNA so low in Fiebig III/IV while p24 as major GAG protein is very much present in Figure 1D. It's crucial to understand the difference between those observations as Fig 2D suggest that patients treated during Fiebig III/V can still be considered as having a limited sanctuary reservoir.

Figure 2F: cell-associated viral loads in LNMCs was measured using a commercial viral load assay Cobas Ampliprep HIV-1 test. Is this commercial kit validated to measure VL in LN? How is DNA contamination excluded (from the same cells)? Why was that not performed by the – for reservoir research often used QPCR assays described by Pasternak et al, or via ddPCR by Kiselina et al or Yukl et al?

Figure 3C: whereas a slight delay in treatment initiation (Fiebig III or later) was associated with significant GCTfh expansion comparable to untreated HIV infection.

Data for Fiebig III/V might be driven by an outlier (higher than late Tx or non Tx). Observations made on very small numbers.

Figure 3E; area density results do show a reduced expansion in late treatment contradicting the results of 3C figure. Moreover also 3C show very limited difference between Fiebig I/II and Fiebig III/V.

The authors write: Overall, these results show that treatment mitigates GCTfh responses : the data is suggestive but not conclusive. It is very difficult to make general conclusions of differences between the Fiebig groups based on the current dataset. One can assume that the Fiebig groups function as one continuum, but comparison of the data do not (always) result in significant differences between I/II and III/V.

Figure 3K: not enough datapoints to draw meaningful conclusions

Figure 4C: how many cells were examined in each subset. Was the row ddPCR data uploaded for reviewing? Was normalisation for input performed on a panel of ref genes (cfr MIQE guidelines)?

Many samples in the CXCR3- population are scoring very low suggesting very low input.

Figure 5 A/B the data are not statistically significant different. Where those assays performed in triplicate. Would need to see the standard deviation.

Discussion

Mitigated GCTfh responses decreased the number of cellular targets of HIV infection:

Hence, GCTfh seem to expand when treatment is not installed. Tfh are also a target for HIV. What is the chicken and what is egg part of that biological observation?

Overall, I do appreciate that differences can be observed between Fiebig I/II and late treatment. Whether differences are apparent between Fiebig I/II and Fiebig III/V is still unclear.

22nd March 2022

Senior Editor
Nature Communications

Re: Baiyegunhi et al, "HIV-1 persistence in lymph node T follicular helper cells (TFH) is mitigated by functional virus-specific T cell responses during hyperacute-treated HIV-1 infection"

Dear Editor,

We greatly appreciate instructive comments from the reviewers. Based on your recommendation, we have performed additional experiments and made substantive changes that have significantly improved the clarity and validity of our findings. As you will see, the new data are mostly consistent with our initial findings.

The reviewers required additional information on the tissue microenvironment, control staining for gp120 by flow cytometry and viral load determinations. We now provide additional data that better elucidate the dynamics of the lymph node microenvironment. Specifically; i) we have quantified the numbers and sizes of GCs per lymph node section, and ii) we have determined the correlative relationship between GCs distribution and p24 and HIV RNA levels in the tissue, iii) we have measured the levels of inflammatory soluble CD14 in plasma and iv) Inducible T cell costimulator (ICOS) receptor expression by GCTfh cells in response to a comment by reviewer 1, iv) we have also performed flow cytometry staining for gp120 using HIV negative samples and *in vitro* infected cells as negative and positive control conditions respectively. The new data in the revised figure 4 clearly demonstrate the specificity of the 3BNC117 antibody binding to HIV infected cells. We are unable to perform HIV RNA quantification using the method suggested by the reviewer due to unavailability of samples, we highlight this as a potential limitation of the study. Moreover, we have toned down conclusions that aren't sufficiently powered by our data, as requested by reviewers #2 and #3.

Overall, the new data have greatly improved the manuscript and we thank the reviewers for the comments and suggested edits. Below is a point by point rebuttal of all the reviewer comments.

Reviewer #1

This reviewer appreciates the importance of the human material used in this study as well as the difficulties for access to lymph node biopsies from relevant tissues. The authors propose that future studies using FNA preparations could be useful at this point. The authors should comment on the advantages and disadvantages of such approach.

The presented cohort is biased regarding the age of the participants. Given the additional lymph node abnormalities found in elderly, the authors should comment on the possible effect of such abnormalities in aging HIV infected subjects.

Answer: We thank the reviewer for acknowledging the significance of our work. We have expatiated on our statement regarding the use of fine needle aspirates by future studies in lines 413 to 419. We have also highlighted the age bias of our study participants as a limitation of our study (lines 419 to 426).

For a better appreciation of the tissue microenvironment on the observed dynamics, a general description (number of active follicles across the groups of donors, size of the active follicles, whether p24 or HIV RNA staining is associated with enlarged, lysed follicles or 'intact/preserved' follicular structures) should be provided.

This is an important comment that the reviewer makes, we already had data that shows the number of active follicles in Fiebig I/II treated (Tx) donors in Fig. 1b and the distribution of p24 within these GCs. Overall our data shows heterogeneity in p24 staining within GCs in the same LN section. We have now quantified the size of active follicles (by BCL-6 staining) across the study groups and also included the number of GCs (new Figs. 3F and S3A). And these data show that the active follicles are smaller in Fiebig I/II Tx participants compared to late Tx and unTx participants.

In addition we have now included new data that show the correlation between p24 and HIV RNA staining with the quantified numbers and sizes of GCs. These data are added as a new supplementary Fig. 3B-E. The new data show a positive relationship between amounts of HIV antigen and the follicular distribution within each tissue. These results have improved the clarity of our findings and we thank the reviewer for the instructive comment.

Previous data have shown a reducing size of the lymph node Tfh cell latent reservoir with respect to time after cART initiation. The authors report no association between Gag p24 staining and treatment duration. Is this the case for the actively transcribed virus CD4 T cells (RNAscope)? If this is the case, the authors should comment on the possible dissociation between latent and 'active' reservoir of the virus.

While Previous data have shown a reducing size of the lymph node Tfh cell latent reservoir with respect to time after cART initiation¹, we observed no association between treatment duration and HIV antigen persistence. These differences may be largely attributed to differences in the study cohort. While we studied individuals who initiated ART in hyper acute HIV, the onset of ART in this other study is later in infection. Their study participants were on therapy for longer periods of time, for up to 12 years. Moreover in that study, the Tfh reservoir correlated with treatment duration while non-Tfh reservoirs did not. Together, differences in study designs may impact the results reported. We have included a comment on active and latent reservoirs in lines 351 to 356 as per the reviewers request.

Figure 1E: the data clearly show two groups of follicles even in the Fiebig I/II donors. Are the high p24 follicles originating from one or several donors? Is this profile associated with any of the other parameters under investigation?

We thank the reviewer for this comment and refer the reviewer back to Fig. 1B that shows p24 density in different follicles within each Fiebig I/II Tx donor. The data shows that 6 out of 14 donors have follicles with high p24 staining, demonstrating that the p24 staining in Fig. 1E is not attributable to just a few individuals. Yes, this profile is associated with RNA staining because we showed a correlation in Fig. 2E, between p24 and HIV RNA densities.

Figure 3C: no difference was found between Fiebig I/II and late Tx donors. Similar profile was found when GC Tfh were calculated from generated images. Most likely this profile reflects the elimination of accumulated Tfh cells upon cART treatment. Still, it would be informative if the authors provide any information related to inflammatory cell types and abnormalities (e.g. fibrosis) in the tissues under investigation.

We agree with the reviewer that chronic inflammation contributes to LN abnormalities, unfortunately, we did not have available tissue samples to measure fibrosis or other tissue abnormalities. We however measured inflammation by quantifying the levels of plasma soluble CD14 in Fiebig I/II Tx (n=6) and Late Tx (n=2) groups. The results in a new supplementary Fig. 3H-I, showed similar sCD14 levels at baseline and after 1 month of infection. These results are inconclusive due to low numbers of patients interrogated. Unfortunately only 2 of our late Tx samples were from a cohort with longitudinal plasma samples, hence the selection of these 2. We also measured ICOS expression as a marker of GCTfh activation. While GCTfh cells had very high ICOS expression compared with nonGCTfh cells, there was a trend of higher ICOS expression in late Tx compared to Fiebig I/II Tx. We have included these results in supplementary Figs. 3F to G.

Figure 3F and text: the authors use the term GC Tfh 17, a term that was first introduced for 'circulating' Tfh cells. However, GC Tfh cells are not capable of producing IL17 even after PMA stimulation. The authors should use this terminology with caution.

We agree with the reviewers have replaced the term GCTfh17 with the phenotypic description of CXCR3-CCR6+ (R6⁺)GCTfh (line 218).

Figure 3: it is known that several CD4 T cells within the follicular areas do not express Tfh-phenotypic markers. What are the numbers of total CD4 T cells in these follicles? What is the ratios of GC Tfh/total follicular CD4 T cells in these follicles?

We thank the reviewer for the instructive comment. High PD-1 expression is a hallmark of Tfh cells so we have quantified PD1 expressing cells in the GC for 11 LN sections based on sample availability. The ratio of CD4⁺ cells to PD-1⁺ cells in the GCs is almost 1 to 1, implying that most CD4⁺ cells co-express PD-1. The data is shown here.

Figure 1: Ratios of CD4 and PD-1 expressing cells within BCL6⁺ GCs quantified using image cytometry

Figure 4D and E: the authors should fix the gating for the intensification of specific cells (the current gates are too low and possibly contaminate the analysis outcome). Although the reviewer believes that the final profile will remain the same, the 'new' data will be more robust.

We have made extensive revisions to Fig. 4 with more control conditions and the new flow plots have adjusted gates.

Lines 281, 283, 307: the text does not reflect what the corresponding figures show for the relevant populations (possibly a typo?)

We have corrected the typo.

Reviewer #2

Figure 1 shows HIV Gag p24 in GCs, but 1F shows that HIV Gag can still be found up to 5-7% of the extra-follicular areas with the differences in GCs really driven by 4 participant samples (whereas the p24 is similar between EF and GC in the other participant samples). While this was statistically significant, it does appear that in a majority of participants, HIV gag was evenly distributed within and without GCs. Representative images of a patient with similar levels (which is the majority) should also be shown in panel D. As of now, it is possible that these figures were selected from the outliers as in panel F. These points should also be addressed more specifically in the results and limitations.

We agree with the reviewer that not all participants had p24 exclusively in the GCs. We have revised the text to indicate that this observation was a trend in some samples (lines 144 and 145).

We have added a supplementary fig 1I showing images with p24 outside GCs. We thank the reviewer for this observation.

The analysis of immediate versus delayed therapy in terms of GCTfh expansion is interesting, but the way the data are presented are misleading. More specifically, the authors show that there is only a trend towards significance between Fiebig I/II and uninfected controls in

terms of cell expansion versus slightly later initiation of ART led to significant differences in expansion between HIV infected and uninfected controls. However, this data does not necessarily support a direct comparison between early and delayed ART initiation and these data are not provided. In order to make the statement that there are differences between HIV treatment groups, a direct comparison is needed (I am assuming it was negative or would have presented in the paper). This data should be shown and indirect comparisons with two groups, and if there is not difference, should be highlighted in the paper.

We thank the reviewer for this observation., we have done direct comparisons as suggested and do not see a statistically significant difference btw Fiebig I/II Tx and Fiebig III-V Tx or Fiebig I/II Tx and late Tx. This is now reflected in lines 199 to 200. However we do see a significant difference between Fiebig I/II Tx and unTx. We have therefore revised the text to read “ Immediate therapy was associated with significant diminution of GCTfh expansion (Fiebig I/II Tx vs unTx $P=0.002$), which was comparable among all treatment groups ($P=ns$, Fig. 3C) [lines 198 to 200].

The tetramer studies to identify HIV-specific CD4 populations is interesting but given only a handful of participant samples were analyzed, it seems difficult to draw a conclusion that the tetramer-specific GCTfh and nonGCTfh were at similar frequencies between untreated and Fiebig I/II, although the authors do acknowledge that there were insufficient quantities of tetramer + cells to determine if antigen specific CD4 T cells were preferentially infected.

We thank the reviewer for acknowledging the significance of the tetramer staining data, but we concede that we do not have a small sample size and we have therefore moved the results to supplementary materials (suppl Fig. 3J-K).

One of the more interesting aspects of this study which seems to be downplayed a bit is the identification of cell that express surface gp120 in tissues in both early and later treated individuals. Whereas most of tissue-based analyses have looked at p24, there is little understanding of HIV env expression in the setting of treated HIV infection. This result is important as it suggests that various immunological modalities (such as CAR-T and bnAbs) may play an important role in reservoir eradication strategies. Figure 4 D and E however use isotype controls for flow staining on LNMC versus PBMC but a more appropriate comparison would be uninfected LNMC and PBMC with 3BNC117. The reason for this is that the low numbers of positive cells both in PBMC and LN could be from non-specific binding of the bNab outside of HIV infection and negative tissue stained with 3BNC is really needed to determine if these cells are expressing HIV gp120.

We thank the reviewer for highlighting the significance of *ex vivo* detection of gp120 positive cells in lymph node samples. Based on the reviewer’s recommendation we have now performed 3BNC117 staining of HIV negative samples as well as cells infected *in vitro* with NL4-3 as controls. The new data highlight the specificity of our flow cytometry staining. The new data are presented in the revised Fig. 4D and described in lines 260 to 261.

I am not sure if SEB stimulation is the appropriate readout for deficiencies of CXCR5+ CD8+ T cells to secrete cytokines. It seems that differences in response to HIV peptide stimulations

are far more relevant to the question at hand. Furthermore, prior work has already clearly demonstrated dichotomies between cytotoxicity of CD8 T cells within LN follicles and cytolytic potential/killing of cells in these compartments. At the least, measures of GzB/Perforin/CD107a etc. could provide a bit more information on HIV-specific responses rather than superantigen response to IFN production alone.

We thank the reviewer for this important observation. We used IFN-g secretion in Fig. 5G as a proxy for HIV-specific cells consistent with previous reports^{2,3}. Our data show fewer HIV-specific cells expressing CXCR5 compared to CXCR5 negative cells. We have also corrected the mislabeled figure 5H (now Fig. 5G) to reflect fewer frequencies of IFN-g + cells among CXCR5+ population. We have removed SEB data as suggested by the reviewer. Please note that the proliferation data in Fig. 5E is a better *in vivo* measure of T cell function⁴.

Fig 5 E is quite interesting but I think there may be a chicken/egg conundrum here. The data show that in cells from low gag density have much more profound proliferation in response to various HIV gag/nef stimulation. As read, this could be simply that CD8 T cell responses with better HIV-specific proliferative capacity leads to reduced reservoir size. Could there also be dysfunctional responses caused by high HIV burden within these tissue regions? In other words, could a lower initial burden of HIV in tissue actually lead to a better, more efficient anti-HIV CTL response than those with much higher burdens of infection?

The reviewer makes a good point and we have incorporated this in our discussion. The new sentence now reads thus:

“It is difficult to determine if the low antigen environment leads to the development of better CD8⁺ T cell function or whether superior CD8⁺ T cell functionality results in lower antigen burden in LN tissue. However, we did not have longitudinal data to unpack this conundrum” (lines 396 to 400).

Of course, longitudinal data would certainly be able to clarify some of the above issues in more detail, but the authors make the valid limitation statement that excisional biopsies are morbid and can't be performed regularly on human participants.

We thank the reviewer for the comment.

The authors state that there is no correlation between circulating and LN HIV RNA, but sample size is somewhat limited and this seems a strong conclusion to make given the relatively small number of patients and heterogeneity in these HIV persistence measures.

We have toned down this conclusion. The sentence now reads thus... **There was no correlation observed between peripheral CD8⁺ T cell responses and the amount of persistent Gag p24 antigen in the LN, suggesting the peripheral responses may not accurately depict HIV persistence in LNs (lines 291 to 294).**

Minor comments: Overall, figure text is rather small on many figures and very difficult to read without amplifying the data plots on the screen. Larger font sizes would dramatically improve readability.

We appreciate the reviewers comment, we have removed some figure panels in Figs. 4 and 5 to supplementary and increased the font size on the figures. We hope that this has improved the clarity of the figure.

Should Mann U-Whitney be Mann Whitney U?

The typo has been corrected.

Could any of the findings observed in this study be secondary to female sex? Perhaps a bit more in discussion would be good.

This point is expatiated on in the discussion section (lines 420 to 426)

unTx is not a standard abbreviation in manuscript text

The abbreviation is now standardized in the manuscript

Reviewer #3

If this women are at such a high risk, why aren't they on PREP? Is PREP offered to this woman?

The participants were offered PrEp according to South African guidelines. The uptake has been good but retention is poor, just like most PrEP programs in South Africa. We have now included this important HIV prevention information in our cohort description in lines 86 to 88 of the manuscript.

Concerning persistence of GAG p24 antigen in GC: Quantitative image analysis of all treated LNs revealed greater area percent of Gag p24 staining in GCs compared to extrafollicular areas of the tissue ($P=0.04$, Fig. 1F). Together, these data demonstrate that early ART initiation in Fiebig stage I/II limits the magnitude of HIV Gag p24 antigen in LNs, but that Gag p24 can persist predominantly in follicular areas even after 4.5 years of fully suppressive treatment.

The data seems to be strong for Fiebig I/II versus the other Fiebig groups. The comparison GC versus EF is much weaker with a $p=0.04$. The data is drive by 4 patient/LN only. So, I do not agree one can make that claim. In addition, I was wondering whether this is only for Fiebig I/II patients, or is this claim still standing when taking into account all fiebig stages.

We agree that our numbers are small we have toned down the statement, we now point out that greater Gag p24 persistence in GCs is just a trend (lines 144 to 145). We have also added a new supplementary Fig. 1I showing some p24 outside GCs. We thank the reviewer for this observation.

Concerning RNA in LN:

Figure 2D: why is the GAG-POL RNA so low in Fiebig III/IV while p24 as major GAG protein is very much present in Figure 1D. It's crucial to understand the difference between those observations as Fig 2D suggest that patients treated during Fiebig III/V can still be considered as having a limited sanctuary reservoir.

The reviewer has made an important observation. We speculate that this could be due to the fact that historical p24 antigens from early acute infection, can persist in the lymphoid tissues for extended periods of time⁵, which is not the case for HIV RNA. Thus the amount of p24 protein may not accurately reflect the amount of ongoing infection.

Figure 2F: cell-associated viral loads in LNMCs was measured using a commercial viral load assay Cobas Ampliprep HIV-1 test. Is this commercial kit validated to measure VL in LN? How is DNA contamination excluded (from the same cells)? Why was that not performed by the – for reservoir research often used QPCR assays described by Pasternak et al, or via ddPCR by Kiselina et al or Yukl et al?

We agree that qPCR might be the better approach but unfortunately we don't have the samples to repeat these experiments using the suggested methods in the publications.

Figure 3C: whereas a slight delay in treatment initiation (Fiebig III or later) was associated with significant GCTfh expansion comparable to untreated HIV infection.

Data for Fiebig III/V might be driven by an outlier (higher than late Tx or non Tx).

Observations made on very small numbers.

Figure 3E; area density results do show a reduced expansion in late treatment contradicting the results of 3C figure. Moreover also 3C show very limited difference between Fiebig I/II and Fiebig III/V.

The authors write: Overall, these results show that treatment mitigates GCTfh responses : the data is suggestive but not conclusive. It is very difficult to make general conclusions of differences between the Fiebig groups based on the current dataset. One can assume that the Fiebig groups function as one continuum, but comparison of the data do not (always) result in significant differences between I/II and III/V.

We thank the reviewer for this observation., we have done direct comparisons as suggested and do not see a statistically significant difference btw Fiebig I/II Tx and Fiebig III-V Tx or Fiebig I/II Tx and late Tx. This is now reflected in lines 199 to 200. However we do see a significant difference between Fiebig I/II Tx and unTx. We have therefore revised the text to read " Immediate therapy was associated with significant diminution of GCTfh expansion (Fiebig I/II Tx vs unTx $P=0.002$), which was comparable among all treatment groups ($P=ns$, Fig. 3C)' [lines 198 to 200].

Figure 3K: not enough datapoints to draw meaning full conclusions

We agree with the reviewer that our tetramer staining data is limited so we've moved it to supplementary (Fig. S3J-K).

Figure 4C: how many cells were examined in each subset. Was the row ddPCR data uploaded for reviewing? Was normalisation for input performed on a panel of ref genes (cfr MIQE guidelines)?

Many samples in the CXCR3- population are scoring very low suggesting very low input.

We now provide raw data as requested by the reviewer. (We would like to clarify that the cxcr3- cells did not have low input as evidenced by the quantified β 2M transcripts from the same well (both the HIV and β 2M probes were included in each well and read using different channels; FAM and VIC).

Figure 5 A/B the data are not statistically significant different. Where those assays performed in triplicate. Would need to see the standard deviation.

No they were not run in triplicates. Each data point represents an individual donor.

Discussion:

Mitigated GCTfh responses decreased the number of cellular targets of HIV infection: Hence, GCTfh seem to expand when treatment is not installed. Tfh are also a target for HIV. What is the chicken and what is egg part of that biological observation?

There appears to be a linear relationship between GCTfh and antigen persistence from our results. High antigen load during infection drives GCTfh expansion⁶ which results in higher target cells for HIV infection.

Overall, I do appreciate that differences can be observed between Fiebig I/II and late treatment. Whether differences are apparent between Fiebig I/II and Fiebig III/V is still unclear.

We agree with the reviewer that the differences between the early Fiebig stages are very subtle and we may require much larger sample sizes to clearly detect differences. This is indicated in lines 427-429 of the discussion.

References

1. Banga, R., *et al.* PD-1(+) and follicular helper T cells are responsible for persistent HIV-1 transcription in treated aviremic individuals. *Nature medicine* **22**, 754-761 (2016).
2. Horton, H., *et al.* Correlation between interferon- gamma secretion and cytotoxicity, in virus-specific memory T cells. *J Infect Dis* **190**, 1692-1696 (2004).
3. Gauduin, M.C., *et al.* Optimization of intracellular cytokine staining for the quantitation of antigen-specific CD4+ T cell responses in rhesus macaques. *J Immunol Methods* **288**, 61-79 (2004).
4. Ndhlovu, Zaza M., *et al.* Magnitude and Kinetics of CD8+ T Cell Activation during Hyperacute HIV Infection Impact Viral Set Point. *Immunity* **43**, 591-604 (2015).

5. Popovic, M., *et al.* Persistence of HIV-1 structural proteins and glycoproteins in lymph nodes of patients under highly active antiretroviral therapy. *Proceedings of the National Academy of Sciences of the United States of America* **102**, 14807-14812 (2005).
6. Lindqvist, M., *et al.* Expansion of HIV-specific T follicular helper cells in chronic HIV infection. *The Journal of clinical investigation* **122**, 3271-3280 (2012).

REVIEWERS' COMMENTS

Reviewer #1 (Remarks to the Author):

The authors have adequately addressed my comments. The revised version can be accepted for publication after a slight modification. I believe the data shown in the supported figure 1 for the reviewer should be included in the suppl material and cited in the text. For several of the donors, the calculated ratio is significantly different from 1, which is expected, at least to this reviewer.

Reviewer #3 (Remarks to the Author):

The authors have substantially revised the manuscript and they provided adequate answers tot the questions. Therefore I consider the manuscript ready for publication

7th June 2022

Senior Editor
Nature Communications

Re: Baiyegunhi et al, "HIV-1 persistence in lymph node T follicular helper cells (TFH) is mitigated by functional virus-specific T cell responses during hyperacute-treated HIV-1 infection"

Dear Editor,

We greatly appreciate instructive comments from the reviewers which have greatly improved the manuscript.

Below is a point by point rebuttal of all the reviewer comments.

REVIEWERS' COMMENTS

Reviewer #1 (Remarks to the Author):

The authors have adequately addressed my comments. The revised version can be accepted for publication after a slight modification. I believe the data shown in the supported figure 1 for the reviewer should be included in the suppl material and cited in the text. For several of the donors, the calculated ratio is significantly different from 1, which is expected, at least to this reviewer.

Answer: We thank the reviewer for acknowledging the significance of our work and for the instructive comment. The figure is now included as supplementary figure 3K and described in the manuscript in line 210.

Reviewer #3 (Remarks to the Author):

The authors have substantially revised the manuscript and they provided adequate answers to the questions. Therefore I consider the manuscript ready for publication.

Answer: We thank the reviewer for acknowledging the significance of our work.